# Current and Future Therapeutic Targets for Directed Molecular Therapies in Cholangiocarcinoma

**DOI:** 10.3390/cancers16091690

**Published:** 2024-04-26

**Authors:** Philipp Heumann, Andreas Albert, Karsten Gülow, Denis Tümen, Martina Müller, Arne Kandulski

**Affiliations:** Department of Internal Medicine I, Gastroenterology, Hepatology, Endocrinology, Rheumatology, and Infectious Diseases University Hospital Regensburg Franz-Josef-Strauß-Allee 11, 93053 Regensburg, Germany

**Keywords:** cholangiocarcinoma, targeted therapy, biliary tract cancer, molecular-directed therapy

## Abstract

**Simple Summary:**

Molecular targeted therapy for cholangiocarcinoma (CCA) involves using drugs that specifically target molecules or pathways involved in molecular pathogenesis. In cholangiocarcinoma, specific mutations in different pathways occur more frequently than in other solid tumors; therefore, specific analysis is recommended early during the course of the disease. Targeted therapies in CCA aim to block the signals that promote cancer cell growth and survival, leading to tumor shrinkage and improved patient outcomes. However, targeted therapies may only be effective in a subset of patients with specific molecular alterations, highlighting the importance of molecular profiling to guide treatment decisions. Additionally, resistance to targeted therapies can develop over time, necessitating ongoing research into novel therapeutic strategies and combination approaches to improve treatment efficacy.

**Abstract:**

We conducted a comprehensive review of the current literature of published data, clinical trials (MEDLINE; ncbi.pubmed.com), congress contributions (asco.org; esmo.org), and active recruiting clinical trains (clinicaltrial.gov) on targeted therapies in cholangiocarcinoma. Palliative treatment regimens were analyzed as well as preoperative and perioperative treatment options. We summarized the current knowledge for each mutation and molecular pathway that is or has been under clinical evaluation and discussed the results on the background of current treatment guidelines. We established and recommended targeted treatment options that already exist for second-line settings, including IDH-, BRAF-, and NTRK-mutated tumors, as well as for FGFR2 fusion, HER2/neu-overexpression, and microsatellite instable tumors. Other options for targeted treatment include EGFR- or VEGF-dependent pathways, which are known to be overexpressed or dysregulated in this cancer type and are currently under clinical investigation. Targeted therapy in CCA is a hallmark of individualized medicine as these therapies aim to specifically block pathways that promote cancer cell growth and survival, leading to tumor shrinkage and improved patient outcomes based on the molecular profile of the tumor.

## 1. Introduction

The incidence of malignant liver tumors has significantly increased in recent decades. Primary liver cancer ranks as the sixth most frequently occurring cancer globally [1]. In 2020, approximately 905.677 new cases of malignant primary liver cancer were reported worldwide, accounting for 4.7% of all diagnosed carcinomas in adults [2]. Among primary liver tumors, hepatocellular carcinoma (HCC) is the most common, comprising 75–85% of cases, followed by intrahepatic cholangiocarcinoma (iCCA) at 10–15% [2]. Less common malignant primary liver tumors include sarcomas and hemangiosarcomas arising from connective tissue or blood vessels, embryonal tumors such as hepatoblastoma, and fibrolamellar carcinoma. iCCA originates from epithelial cells lining the bile ducts (cholangiocytes) within the hepatic parenchyma [3].

The mortality rate associated with primary liver tumors is high, ranking second globally (830.180 deaths per year) after lung tumors, or in third place when considering colon and rectal carcinomas together [2]. The poor prognosis is evidenced by nearly equivalent incidence and mortality rates. Cholangiocarcinoma (CCA) encompasses a heterogeneous tumor entity. Extrahepatic cholangiocarcinoma (eCCA), distinguished from iCCA (classified as primary liver tumors), is further categorized into perihilar (proximal eCCA) and distal types (distal eCCA) based on anatomical localization [4]. Proximal eCCA occurs between second-order bile ducts and the insertion of the cystic duct with the common bile duct, while distal eCCA is confined to the common bile duct below the insertion of the cystic duct. This differentiation holds significance due to different risk factors, molecular and clinical characteristics, and different therapeutic approaches [5]. Proximal eCCA accounts for 50–60% of CCA in the USA, followed by distal eCCA at 20–30% and iCCA at 10% [6,7]. Incidence and mortality data for eCCA vary significantly by region.

Gallbladder carcinoma (GB) represents another tumor entity of the hepatobiliary system. In 2020, 115.949 cases of GB were diagnosed worldwide, with a high associated mortality rate of 84.695 deaths annually [2]. Primary liver cancer (HCC, iCCA), extrahepatic cholangiocarcinoma (eCCA), and gallbladder cancer (GB) collectively present a global health challenge. The absolute 5-year survival rate for CCA ranges between 7% and 20%, contingent on localization, age, and gender [6].

Surgical resection remains the sole curative therapeutic option, albeit with frequent recurrences. Consequently, adjuvant therapeutic concepts have been established and recommended by international guidelines for CCA treatment [8]. Early symptoms of CCA are often absent, leading to late-stage diagnoses where surgical resection is not an option. In advanced stages, systemic oncological therapy serves as the standard of care. Traditional cytotoxic chemotherapy has been historically favored in CCA treatment. Of clinical relevance, recent guidelines recommend combination therapies involving immunotherapeutic agents as the first-line treatment [9].

Furthermore, molecular diagnostics have been established, particularly for CCA therapy aimed at identifying tailored therapeutic strategies. To summarize, targeted therapeutic concepts have been introduced and recommended by international guidelines early in the therapeutic sequence [9]. Due to the rising global incidence of CCA, deeper comprehension of this tumor entity and its pathophysiology is crucial for the development of novel therapeutic strategies and the enhancement of patient prognosis.

The crosstalk regarding molecular targets in intrahepatic and extrahepatic cholangiocarcinoma (CCA) and gallbladder cancer (GBC) is a multifaceted area of research with significant implications for diagnosis and treatment. While these cancers arise from different anatomical sites within the biliary tract, they share some common molecular pathways and genetic alterations, as well as differences that reflect their distinct clinical behaviors and therapeutic responses.

This review aims to offer an overview of established molecular therapeutic approaches in CCA and provide insight into upcoming agents for the near future.

## 2. Risk Factors and Diagnostic Work-Up

Chronic inflammatory activity in the biliary tract stands as the main risk factor for the development of both iCCA and eCCA. Furthermore, various known risk factors, detailed in Table 1, can individually contribute to sustained inflammation. PSC constitutes the major inflammatory condition in daily clinical practice.

The primary risk factor contributing to the development of GB is chronic inflammation, alongside additional factors such as gallbladder stones, obesity, and gallbladder polyps (refer to Table 2).

Early clinical symptoms are typically absent in CCA. Patients often initially present with nonspecific symptoms like fatigue, exhaustion, weight loss, loss of appetite, and nonspecific abdominal pain. Symptoms of cholestasis, a common complication of biliary obstruction, usually appear in later stages and may include fever, jaundice, itching/pruritus, discolored stool, and dark brown urine. Notably, CCA is incidentally discovered in 20–30% of cases during abdominal imaging [57].

For suspected CCA based on clinical symptoms, initial diagnostic steps should involve sonography (including contrast-enhanced ultrasound (CEUS)) and contrast-enhanced computed tomography (CT) of the abdomen [58,59]. CEUS, as well as contrast-enhanced CT and MRI, are particularly valuable in distinguishing CCA from HCC [60]. In CT diagnostics, a three-phase cross-sectional imaging protocol (arterial phase, portal venous phase, delayed phase) is recommended [61]. Similarly, a three-phase imaging approach is advisable when incidentally detecting a hepatic mass on CT for staging purposes. CT and CEUS both provide an impression of biliary obstruction and delineate vascular anatomy prior to surgical resection [61,62]. CCA often exhibits progressive contrast uptake across different phases, while HCC typically displays early arterial hypervascularization followed by washout in subsequent phases [63,64,65]. GB diagnosis is frequently established incidentally during cholecystectomy for cholecystolithiasis or cholecystitis. Therefore, the resection of incidental GB should be followed by appropriate staging. If the suspicion of a hepatobiliary system tumor persists despite the absence of a tumor mass in cross-sectional imaging, endoscopic retrograde cholangiography (ERC), complemented by cholangioscopy (CC), or an endoscopic ultrasound (EUS) should be performed for diagnosis and staging. Histological confirmation via brush cytology or forceps biopsy (ERC and cholangioscopy), as well as fine-needle aspiration cytology (EUS), can enhance specificity, albeit with modest sensitivity [66,67]. Sensitivity in diagnosing patients with CCA can be increased further by using CC [68].

Prior to surgical resection, staging MRI/magnetic resonance cholangiopancreatography (MRCP) is recommended, particularly for assessing local tumor spread along the bile ducts, where MRI outperforms CT imaging [69]. PET-CT scans are recommended when distant metastases, lymph node involvement, or tumor recurrence are suspected [70]. The tumor marker carbohydrate antigen 19-9 (CA 19-9) plays a limited role in confirming the diagnosis due to elevations also occurring in benign cholestatic diseases. Of clinical relevance, levels of CA 19-9 above 1000 U/mL suggest the presence of CCA [71]. Furthermore, CA19-9 can be utilized for post-resection follow-up or as a monitoring parameter during therapy if initially elevated. Given the absence of distinctive positive diagnostic imaging criteria, histological confirmation of CCA is typically necessary. However, in cases where resectability is probable and there is a high suspicion of intrahepatic or extrahepatic bile duct carcinoma, preoperative biopsy confirmation may be bypassed in favor of examining the entire tumor tissue post-resection. An extensive diagnostic and therapeutic plan should be developed in an interdisciplinary tumor board. The diagnostic algorithm for cholangiocellular carcinoma is shown in Figure 1. For unresectable cases of intrahepatic and extrahepatic cholangiocarcinoma (iCCA, eCCA), and gallbladder cancer (GB), histological confirmation before starting systemic therapy is crucial. Additionally, ensuring a sufficiently sizable biopsy is imperative to yield ample tissue for subsequent molecular pathological diagnostics [72].

Noncoding RNA (ncRNA) has garnered increasing attention in recent years as a potential biomarker for various diseases and conditions. Unlike coding RNA, which serves as a template for protein synthesis, ncRNA does not encode proteins but plays crucial regulatory roles in gene expression, cellular processes, and disease pathways. ncRNAs are remarkably stable in bodily fluids such as blood or urine. This stability is due to their encapsulation in extracellular vesicles, association with proteins, or protection from degradation by ribonucleases. Because of this stability, ncRNAs can be readily detected and quantified in clinical samples, making them attractive biomarker candidates. Non-coding RNAs (ncRNAs) include microRNA (miRNA), long non-coding RNA (lncRNA), and circular RNA (circRNA) [73]. Some studies have shown that tumor cells and bile fluid in patients with CCA exhibited an increased expression of long non-coding RNA (lncRNA). In one of these studies, it was discovered that two lncRNAs, ENST00000588480.1 and ENST00000517758.1, exhibited high expression levels in bile-derived exosomes of CCA patients. Combining these two lncRNAs for diagnostic purposes resulted in an area under the curve (AUC) of 0.709, with sensitivity and specificity values of 82.9% and 58.9%, respectively. The sensitivity of these lncRNAs surpassed that of serum CA19-9 (82.9% vs. 74.3%). Furthermore, it was verified that higher expression levels of these two lncRNAs in CCA patients correlated with poorer survival outcomes, indicating their potential utility as prognostic markers for monitoring CCA [73,74]. Another study reported the downregulation of lncRNA-NEF in the plasma of iCCA patients, which served as a promising diagnostic marker, effectively distinguishing iCCA patients from healthy controls. However, the study did not provide information on the sensitivity and specificity of lncRNA-NEF as a diagnostic marker for iCCA. Furthermore, iCCA patients with low lncRNA-NEF expression exhibited significantly lower overall survival (OS) rates (*p* = 0.0198) [75]. The ncRNA DLEU1 was linked to advanced lymph node infiltration in CCA patients, correlating with poorer OS in those with high DLEU1 expression. Notably, DLEU1 emerged as a valuable prognostic marker for CCA, aiding in predicting CCA prognosis [76]. The significance of circular RNA (circRNA) has also been investigated in studies. For circRNA Cdr1as and circa-CCAC1, their value as both diagnostic and prognostic markers could be demonstrated [77,78]. Furthermore, several other studies confirmed that high expression levels of lncRNA (PSMA3-AS1, ZEB1-AS1, HOXD-AS1, SNHG20, LINC00667, FOXD2-AS1) were prognostic markers for iCCA [79,80,81,82,83,84]. Several other prognostic-related lncRNAs in cholangiocarcinoma are reported [85].

Despite their potential, several challenges need to be addressed before ncRNAs can be widely adopted as biomarkers in clinical practice. These include the need for large-scale validation studies, standardization of detection methods and data analysis pipelines, elucidation of functional roles and underlying mechanisms, and development of robust assays suitable for routine clinical use. Additionally, factors such as sample heterogeneity, inter-individual variability, and technical limitations may affect the accuracy and reliability of ncRNA-based biomarker assays. As with any biomarker or diagnostic test, ethical considerations regarding patient consent, privacy, and data sharing must be carefully addressed.

## 3. Stage-Dependent Therapeutic Regimes

The therapeutic approach for CCA depends on tumor localization and the extent of spread. An overview of the currently recommended therapy for cholangiocarcinoma is shown in Figure 2. Presently, surgical resection aiming for R0 status remains the sole curative option, provided there is no distant metastasis. Prognostic factors influencing tumor recurrence post-resection include lymph node involvement, vascular invasion, and multifocal tumor localization [72,86,87,88]. Patients with early-stage tumors benefit from adjuvant systemic therapy following resection, as per international guidelines recommending a 6-month course of capecitabine (BILCAP study) [9,72,89,90].

For primarily unresectable locally advanced tumors, neoadjuvant systemic therapy is currently being tested in prospective, randomized trials. Chemotherapy, analogous to palliative treatment, typically involves gemcitabine plus cisplatin, with the addition of durvalumab or pembrolizumab if suitable for immune checkpoint blockades, as advised by current international guidelines [9,72,91,92]. Based on the data from the TOPAZ-1 study, durvalumab was approved by the FDA in September 2022 and by the EMA in December 2022. Overall survival was the primary endpoint of TOPAZ-1. The combination of gemcitabine/cisplatin with durvalumab significantly prolonged overall survival compared to chemotherapy alone (HR 0.77; *p* = 0.0008). Durvalumab increased the response rate to 24.4% compared to 17.1% in the standard arm. Progression-free survival was also significantly prolonged, at 5.7 vs. 7.2 months (HR 0.76; *p* = 0.0005). The mOS was 12.8 months (95% CI, 11.1–14.0) in the durvalumab group and 11.5 months (95% CI, 10.1–12.5) in the placebo treatment group [91]. Recently, the KEYNOTE-966 trial showed positive results for the combination of pembrolizumab with gemcitabine and cisplatin in the palliative setting. Patients in the pembrolizumab group showed an mOS of 12.7 months compared with 10.9 months among those in the placebo arm. The mPFS was 6.5 months for pembrolizumab plus gemcitabine/cisplatin and 5.6 months for the standard-of-care gemcitabine/cisplatin [93]. Pembrolizumab in combination with gemcitabine/cisplatin has also been approved by the FDA and EMA for the first-line treatment of CCA in palliative settings.

It is worth noting that GB did not benefit from the addition of immunotherapy in either study. Furthermore, PD-L1 expression was not found to be a predictive marker for antitumor activity. In the TOPAZ-1 study, patients with iCCA and eCCA showed a comparable benefit in terms of mOS (HR (95% CI), 0.76 (0.58–0.98), respectively, 0.76 (0.49–1.19)). Nevertheless, patients with GB showed only a moderate benefit when adding durvalumab (HR (95% CI), 0.94 (0.65–1.37)) [91]. Consistent with these data, the iCCA in the KEYNOTE-966 study benefited the most in terms of mOS from the addition of pembrolizumab (HR (95% CI), 0.76 (0.64–0.91)), while eCCA and GB showed only modest benefits regarding mOS (HR (95% CI), 0.99 (0.73–1.35), respectively, 0.96 (0.73–1.26)) [93]. The extent to which treatment recommendations can be derived from these data for the shifting subtypes in everyday clinical practice remains to be seen in the forthcoming results in real-world studies. The outcomes of patients treated in the gemcitabine and cisplatin groups of TOPAZ-1 and KEYNOTE-966 were comparable to historical controls of gemcitabine and cisplatin. Long-term survival data from both studies are pending in order to conclusively evaluate the long-term effect of adding immunotherapy to gemcitabine/cisplatin. In both trials, caution is advised when interpreting subgroup analyses as they were not adjusted for multiplicity, and the trials were not adequately powered to compare outcomes in individual subgroups. Both gemcitabine/cisplatin in combination with durvalumab and pembrolizumab showed incidences of treatment-related adverse events of CTC grades 3 and 4 that were comparable to those seen with treatment using gemcitabine/cisplatin. Notably, regarding treatment design, disparities were evident between the two trials. In TOPAZ-1, gemcitabine was restricted to eight cycles, whereas in KEYNOTE-966, gemcitabine administration could continue until disease progression or intolerable toxicity, without a specified maximum number of cycles. The survival data of both studies for the special subgroups of patients with chronic viral hepatitis (B and C) have not yet been published. The IMbrave 150 study on the use of atezolizumab/bevacizumab in patients with hepatocellular carcinoma showed an advantage of immunotherapy in the subgroups of patients with viral genie. It remains to be seen to what extent patients with viral hepatitis will benefit from immunotherapy in the CCA. For second-line treatment, there is clinical evidence for the efficacy of FOLFOX (ABC-06 study). The ABC-06 study showed a benefit for FOLFOX versus active symptom control for advanced biliary tract cancer [94]. The overall survival was significantly longer in the FOLFOX group than in the active surveillance group, with an mOS of 6.2 months (95% CI, 5.4–7.6) in the FOLFOX group versus 5.3 months (95% CI, 4.1–5.8) in the active surveillance group (adjusted HR 0.69). Another second-line option is an irinotecan-based chemotherapy. For the NALIRI protocol (liposomal irinotecan plus fluorouracil and leucovorin), there exists divergent clinical evidence for anti-tumor activity. The NIFTY trial (phase IIb, NCT03524508), in their extended follow-up report (median 33.2 months), reported a significant improvement in mPFS (independent central review) for NALIRI compared with 5-FU for patients with advanced biliary tract cancer in an Asian patient population after pre-treatment with gemcitabine/cisplatin (mPFS: 4.2 months, 95% CI, 2.8–5.3 vs. 1.7 months, 95% CI, 1.4–2.6) [95,96]. The mOS was 8.6 months (95% CI, 5.4–10.5) in the nanoliposomal irinotecan group compared to 5.3 months (95% CI, 4.7–7.2) in the control group [96]. Controversially, the NALIRICC trial (phase II, NCT03043547) did not demonstrate a benefit for NALIRI compared with 5-FU. The reported mOS was 6.9 months for NALIRI compared with 8.21 months for 5-FU [97]. The value of NALIRI in second-line therapy, therefore, remains unclear. For patients with molecular druggable alterations and approved molecular-directed therapeutic options, targeted therapy should be considered in second-line therapy, as impressive survival rates have been achieved here; see the section below for details.

The reassessment of resectability after 2–3 months of systemic therapy is crucial, and resection should be discussed in case of a good response. Liver transplantation under specific criteria (Mayo criteria) might be considered if there are no distant metastases, no lymph node involvement, and the criteria are met [98,99]. Mayo criteria include a tumor diameter < 3 cm, the absence of lymph node metastases, extrahepatic tumor manifestation, histologic evidence of proximal eCCA, and the elevation of tumor marker CA19-9 > 100 U/mL with radiologic evidence of malignant stenosis [100,101]. In the palliative setting with local unresectability or distant metastasis, commencing oncologic systemic therapy is recommended [9,72]. Molecular characterization of the tumor and consultation with molecular NGS sequencing should also be performed in a palliative setting [9]. Recent studies exploring targeted drugs have shown promise, with up to 40% of patients with CCA exhibiting druggable genetic alterations in studies like MOSCATO-01 [102]. Participation in ongoing clinical trials, following discussions at an interdisciplinary tumor board, should be considered at all stages of treatment.

## 4. Current and Future Molecular-Directed Therapeutic Agents in CCA

### 4.1. Prevalence of Druggable Molecular Targets in Different Subtypes of CCA

Molecular analyses in patients with CCA have revealed a large number of addressable target structures (see Table 3). Throughout the remainder of the review, the current evidence, as well as planned, initiated, and ongoing studies, are reported and discussed. The NGS analyses also detected different incidences of treatable mutations for the various CCA entities (see Table 3).

Currently planned, initiated, and recruiting clinical studies on the specific molecular targets are listed in their respective sections (see Section 4.2, Section 4.3, Section 4.4, Section 4.5, Section 4.6, Section 4.7, Section 4.8, Section 4.9, Section 4.10, Section 4.11, Section 4.12, Section 4.13, Section 4.14, Section 4.15, Section 4.16, Section 4.17, Section 4.18, Section 4.19). The distribution of druggable molecular targets among the CCA subtypes varies for specific alterations, as illustrated in Figure 3.

The distribution of molecular alterations varies greatly between the different CCA subtypes; nevertheless, there is some overlap in the downstream signaling cascades for the different targets. FGFR2 alterations, which tend to cluster in iCCA, and HER2 overexpression, which is frequently observed in eCCA and GB, share common downstream signaling components, such as the Ras/Raf/MEK/ERK pathway, the PI3K/AKT/mTOR pathway, and the JAK/STAT pathway [106,107]. Dysregulation of the EGFR signaling cascade plays an important role in the carcinogenesis of all three subtypes of CCA (eCCA, iCCA, GB). The EGFR signaling pathway is involved in regulating various cellular processes such as cell proliferation, survival, differentiation, and migration. The cellular effects are achieved after receptor activation via downstream signaling pathways, such as the Ras/Raf/MEK/ERK pathway, the PI3K/AKT/mTOR pathway, and the JAK/STAT pathway [105]. NTRK fusion cholangiocarcinoma refers to a subset of cholangiocarcinoma (CCA) cases characterized by the presence of fusions involving the neurotrophic receptor tyrosine kinase (NTRK) genes. The constitutively activated NTRK fusion protein phosphorylates various downstream signaling molecules, leading to the activation of multiple intracellular signaling pathways like Ras/Raf/MEK/ERK pathway, PI3K/AKT/mTOR pathway, and PLCγ (phospholipase C gamma) pathway [108]. Molecular alterations within the Ras/Raf/MEK/ERK pathway or the PI3K/AKT/mTOR pathway, such as mutations of KRAS or PIK3CA, are present in all CCA subtypes, where KRAS mutations show the highest frequency in eCCA. In summary, different molecular alterations lead to similar cellular processes such as proliferation, migration, and differentiation via convergent downstream signaling pathways, which contribute to tumorigenesis.

Alterations in genes that are related to the control of DNA damage response (DDR) and cell cycle regulation are also frequent in all CCA subtypes. Mutations of BAP1, PTEN, and PBRM1 are predominantly found in iCCA, whereas mutations of TP53, ARID1A, CDKN2A, CHK1/2, ATM, ATR, and BRCA occur in all CCA subtypes [103,104,105]. DDR pathways encompass a network of signaling cascades and effector mechanisms that maintain genomic integrity and ensure proper cellular responses to DNA damage. Common DDR pathways include the DNA damage sensors (ATM and ATR), signaling kinases (CHK1 and CHK2), DNA repair mechanisms (ARID1A, BRCA, and BAP1), cell cycle regulators (CDKN2A and TP53), and apoptosis induction [109].

Furthermore, there exists a potential crosstalk between BRCA mutations and IDH mutations (predominantly in iCCA) in cholangiocarcinoma. BRCA mutations and IDH mutations can independently affect DDR pathways but through different mechanisms. BRCA1 and BRCA2 are involved in homologous recombination-mediated DNA repair. Mutations in these genes increase genomic instability, leading to a higher susceptibility to DNA damage and potential therapeutic vulnerabilities. IDH mutations result in the production of the oncometabolite 2-hydroxyglutarate (2-HG), which inhibits DNA demethylases and alters epigenetic regulation. This may affect DDR indirectly by influencing chromatin structure and DNA repair gene expression. Inhibiting PARP1 in BRCA-mutated CCA can cause a prolonged presence of single-strand breaks, potentially disrupting replication forks and generating double-strand breaks. IDH1/2 mutations have been shown to heighten sensitivity to PARP inhibitors [109,110].

Overall, the crosstalk between genes related to DDR in cholangiocarcinoma represents a complex interplay between genomic instability and tumor biology. Further research is needed to elucidate the precise molecular mechanisms underlying this crosstalk and its implications for cancer development and treatment strategies.

Approved molecular-directed drugs (FDA) for the treatment of advanced CCA in the palliative setting are FGFR inhibitors (pemigatinib, futibatinib, infigratinib), NTRK inhibitors (larotrectinib, entrectinib), the IDH inhibitor ivosidenib, and checkpoint inhibitors (pembrolizumab and durvalumab in combination with gemcitabine/cisplatin). The signaling pathways involved and the mechanism of action are shown in Figure 4. Furthermore, substances targeting BRAF alterations (a combination of dabrafenib and trametinib), HER2-signaling pathways (trastuzumab deruxtecan), and alterations in the receptor tyrosine kinase RET (selpercatinib) are FDA-approved for treating solid tumors harboring corresponding alterations, including patients with advanced CCA. Due to the increasing implementation of next-gen sequencing in the diagnostics of CCA, the understanding of molecular alterations that contribute to CCA tumor neogenesis will steadily improve in the future and, thus, advance the understanding of the signal transduction pathways involved as well as the development of further targeted drugs.

### 4.2. Target Fibroblast Growth Factor Receptor (FGFR)

The fibroblast growth factor receptors (FGFR) constitute a receptor family binding secreted fibroblast growth factor (FGF). Typically, FGFRs consist of an extracellular domain (comprising three immunoglobulin-like domains), a transmembrane domain, and an intracellular domain with receptor tyrosine kinase (RTK) activity [111]. Among humans, the FGFR family encompasses four active members: FGFR1 to FGFR4, with FGFR5 being a fifth member lacking intracellular RTK activity [112]. Upon binding of FGF, the receptor undergoes dimerization, catalytic autophosphorylation, and subsequent controlled pathway activation via a downstream-regulated intracellular signaling cascade [111].

In CCA, several molecular alterations lead to ligand-independent FGFR2 activation, increased ligand affinity, or disruption of the autoinhibited configuration of the intracellular RTK [113]. Primarily, two genetic rearrangement mechanisms are described: loss of the C-terminal region and gain of domains due to gene fusion (C-terminal fusion), both enhancing receptor dimerization [114]. Enhanced FGF-FGFR signaling exhibits oncogenic roles in CCA [113]. Increased activation of FGF-FGFR signaling causes tumorigenesis by promoting cellular proliferation, migration, survival, invasion, and angiogenesis [115]. The main downstream signaling pathways affected by FGF-FGFR activation include the Ras/Raf/MEK/ERK pathway, the PI3K/AKT/mTOR pathway, and the JAK/STAT pathway [106]. Notably, FGFR2 fusions and rearrangements are predominantly observed in intrahepatic cholangiocarcinoma (iCCA) and occur in 10–20% of patients [116,117,118,119].

Pemigatinib stands as a selective competitive inhibitor of FGFR1, FGFR2, and FGFR3 [120]. In the FIGHT-202 trial (NCT02924376, phase II), 146 patients were enrolled across three cohorts: A, comprising FGFR2 gene rearrangements/fusions; B, other FGF/FGFR gene alterations; and C, no FGF/FGFR gene alterations. In cohort A, the median progression-free survival (mPFS) and median overall survival (mOS) were 6.9 months (95% CI, 6.2–9.6) and 21.1 months (14.8–not reached) respectively. The overall response rate (ORR) in cohort A was 35.5% (95% CI, 26.5–45.4). Conversely, cohorts B and C exhibited no responses [121]. A post-hoc analysis of FIGHT-202 data suggests that patients with CCA and FGFR2 fusions or rearrangements treated with second-line pemigatinib might experience longer PFS compared to classic cytostatic systemic therapy [122]. Pemigatinib has received approval from by the U.S. Food and Drug Administration (FDA) and European Medicines Agency (EMA) for treating previously treated, locally advanced, or metastatic CCA with FGFR2 gene rearrangements/fusions based on positive outcomes from the FIGHT-202 trial. The recommended dosage is 13.5 mg of pemigatinib daily for 14 days, followed by one week off of therapy (21-day cycles). Numerous ongoing clinical studies are assessing the application of pemigatinib in various clinical scenarios in CCA, including adjuvant settings or as a first-line palliative treatment (see Table 4).

Futibatinib is a highly selective, irreversible inhibitor targeting FGFR1, FGFR2, FGFR3, and FGFR4 [123]. Initially evaluated in the FOENIX-101 trial (NCT0205277, phase I, dose-escalation trial), which enrolled 86 patients with previously treated advanced malignancies. After dose-finding, 20 mg futibatinib once daily was established as the recommended dose for the subsequent FOENIX-CCA2 trial (NCT02052778, phase II). This phase II enrolled 103 patients diagnosed with unresectable or metastatic FGFR2 fusion-positive or FGFR2 rearrangement-positive iCCA and experiencing disease progression after one or more prior lines of systemic therapy [124]. Patients received oral futibatinib at a dosage of 20 mg once daily. Across the entire collective of FOENIX-CCA2, 43 patients (95% CI, 32–52) demonstrated tumor response, with a median duration of response (mDOR) of 9.7 months. The mPFS was 9.0 months and the mOS was 21.7 months after a median follow-up of 17.1 months [124]. Subsequently, futibatinib gained approval from the FDA and EMA for treating previously treated, locally advanced, or metastatic CCA with FGFR2 gene rearrangements/fusions. A recent analysis conducted after a median follow-up of 25 months demonstrated a mDOR of 9.5 months. In this updated analysis, the mPFS was 8.9 months, and the mOS was 20.0 months [125]. Ongoing studies for futibatinib are mentioned in see Table 5.

Infigratinib is a selective competitive and reversible inhibitor of FGFR1, FGFR2, and FGFR3 [126]. Infigratinib demonstrated efficacy in previously treated, unresectable locally advanced, or metastatic CCA with FGFR2 fusion or rearrangement in the CBGJ398X2204 trial (NCT02150967, phase II). The trial enrolled 108 patients who received infigratinib orally at 125 mg once daily for 21 days followed by one week off of therapy (28-day cycles). After a median follow-up of 10.6 months, the ORR was reported at 23.1% (95% CI, 15.6–32.2) and the mDOR was 5 months (95% CI, 3.7–9.3) [127]. Based on these results, infigratinib received FDA approval for the treatment of previously treated, locally advanced, or metastatic CCA with FGFR2 gene rearrangements/fusions. Ongoing studies for infigratinib are mentioned in see Table 6.

Derazantinib, a multi-kinase inhibitor with reversible inhibition, exhibits potent blocking activity against FGFR1, FGFR2, and FGFR3. It also targets receptor tyrosine kinase RET (RET), discoidin domain-containing receptor 2 (DDR2), receptor of vascular endothelial growth factor (VEGFR) 1, and receptor tyrosine kinase KIT (KIT) [128].

A phase I/II trial (NCT01752920) investigated the efficacy of derazantinib in patients with unresectable and previously treated CCA with FGFR2 fusion. The study enrolled 29 patients who received derazantinib orally at 300 mg once daily. After a median follow-up of 20 months, the ORR was 20.7%, with a mDOR at 5.8mo and an mPFS of 5.7mo (95% CI; 4.04–9.2) [129]. Data from cohort 2 of the FIDES-01 trial (NCT03230318, phase II) demonstrated an mPFS of 7.8 months (95% CI, 5.5–8.3), with an ORR of 6.8% (95% CI, 1.4–18.7 ) and a DCR of 63.6% (95% CI, 47.8–77.6) [130].

While derazantinib is currently not approved for the treatment of CCA with FGFR2 fusion, data for intrahepatic CCA (iCCA) with FGFR2 mutation/amplification are available. Data from cohort 1 of the FIDES-01 trial (NCT03230318, phase II) reported that derazantinib provides clinical benefit in the treatment of patients with iCCA and FGFR2 mutation or FGFR2 amplification. In cohort 1 of the FIDES-01 trial, 44 patients were enrolled with FGFR2 mutation including missense point mutations (78%) and other short variants (11%) and FGFR2 amplifications (11%). The mPFS was 8.3 months (95%CI, 3.5–16.7), and an ORR of 22.3% (95% CI, 14.7–31.6) and disease control rate (DCR) of 75.7% (95% CI, 66.3–83.6) were reported [130]. Presently, derazantinib lacks approval for treating cholangiocarcinoma with FGFR2 mutation or amplification. Ongoing studies for derazantinib are mentioned in see Table 7.

Erdafitinib exhibits potent tyrosine kinase inhibitory activity against FGFR1-4 with reversible inhibition. Alongside FGFR, erdafitinib also inhibits other tyrosine kinases such as RET, platelet-derived growth factor receptor (PDGFR) A and B, fms-related tyrosine kinase 4 (FLT4), KIT, and VEGFR 2 [131]. In a phase II study (NCT02699606, focusing on the Asian population) patients with FGFR alterations who had previously failed at least one systemic treatment were treated with erdafitinib at a dosage of 8 mg orally once daily. Results showed an impressive ORR of 50%, a DCR of 83.3%, and an mPFS of 5.59 months (95% CI, 1.87–13.67) [132]. However, despite these promising results, erdafitinib currently lacks approval for the treatment of CCA.

Ponatinib, a multi-targeted TKI that includes inhibition of FGFR, was part of a pilot study (NCT02265341, phase II) involving patients with advanced CCA and FGFR alterations. The study was terminated prematurely and out of the 12 enrolled patients, only one showed a partial tumor response [133].

Debio is an oral TKI with high selectivity for FGFR1–3, exerting reversible inhibition [134]. In the expansion phase of a basket trial (NCT01948297, phase I) encompassing solid tumors with FGFR1-3 gene alterations, 5 patients with CCA received Debio (80 mg once daily). Among patients exhibiting FGFR2 fusion, two showed stable tumor disease, and two demonstrated a partial tumor response. Unfortunately, one patient with a FGFR1 fusion did not respond to treatment and showed progressive disease [135]. Further studies are necessary to thoroughly assess the efficacy of Debio in patients with CCA and FGFR gene alterations. The results of the phase II basket trial (FUZE, NCT03834220) are eagerly awaited to provide more insights.

The currently available FGFR2 inhibitors lack high selectivity for the FGFR2 receptor, leading to a broader spectrum of side effects. Additionally, the emergence of FGFR2 resistance mutations poses a significant challenge. RLY-4008 marks the first highly selective inhibitor of FGFR2 and is currently under investigation in the ReFocus trial (phase I/II, NCT04526106). Preliminary data from 17 patients who received the recommended phase II dose demonstrated potent efficacy, with an impressive ORR of 88% (95% CI, 63.6–98.5) [136]. Moreover, tinengotinib has demonstrated potent inhibition of acquired resistant mutations in early clinical studies conducted in CCA [137]. For ongoing clinical trials investigating next-generation FGFR2 inhibitors see Table 8.

### 4.3. Target Isocitrate Dehydrogenase (IDH) 1/2

Isocitrate dehydrogenase (IDH) 1 and 2 catalyze the oxidative decarboxylation of isocitrate to α-ketoglutarate. IDH mutations (heterozygous point mutations) result in elevated levels of 2-hydroxyglutarate. 2-hydroxyglutarate is an oncometabolite that leads to epigenetic changes and abnormalities in cell differentiation, growth factor dependence, or hypoxia signaling [138]. IDH mutations occur frequently in iCCA, less so in GB or eCCA [139,140]. Approximately 15–20% of iCCA patients showed IDH 1/2 mutations [141]. IDH1 mutations are more frequent than those of IDH2 [142].

Ivosidenib demonstrated promising anti-tumor activity in patients with locally advanced or metastatic CCA carrying IDH1 mutation [143]. Ivosidenib is a small-molecule inhibitor of mutant IDH1. In a phase III study (NCT02989857, ClarIDHy Trial), patients were randomly assigned to receive either ivosidenib (500 mg once daily) or a placebo, with crossover from placebo to ivosidenib being allowed. Adjusted for crossover, the mOS with placebo was 5.1 months (95% CI, 3.8–7.6) compared to 10.3 months (95% CI, 7.8–12.4) for ivosidenib. Due to the statistically significant benefit of ivosidenib over placebo, the FDA granted approval for advanced cholangiocarcinoma with IDH1 mutation. In 2023, ivosidenib was also approved by the EMA. Clinical studies addressing treatment with ivosidenib for patients with CCA are reported in Table 9.

Several small molecules are currently being investigated to target IDH1 mutations, IDH2 mutations, or pan-IDH1/2 mutations in several phase I and II trials (see Table 10). In a completed phase II trial involving dasatinib (NCT02428855) among patients with advanced or metastatic CCA and IDH1/2 mutations, the mPFS was 8.7 months, while the mOS was 37.9 months. Results from a phase I/II trial for olutasidenib (NCT03684811) and from a phase I/II trial for enasidenib (NCT02273739) among patients with advanced or metastatic CCA and respective IDH1 or IDH2 mutations are pending. The IDH1/2 inhibitor LY3410738 demonstrated a favorable safety profile in patients with IDH1/2 mutations in advanced solid tumors (NCT04521686) [144]. LY3410738 binds covalently to the mutant IDH1 enzyme at a different site compared to ivosidenib, thereby reducing the risk of secondary mutations [145].

### 4.4. Target Human Epidermal Growth Factor Receptor (HER) 2

HER2 overexpression is frequently observed in CCA, particularly in eCCA, with a prevalence ranging between 5 and 20% [146,147,148]. HER2, an RTK, belongs to a family of four human epidermal growth factor receptors (HER1-4). Dimerization activates HER2, either through homodimerization or heterodimerization with HER1/EGFR1 or HER3. This activation triggers the phosphorylation of tyrosine kinases, leading to cell growth, cell proliferation, and malignant transformation. Several downstream signaling pathways are involved, such as the mitogen-activated protein kinase (MAPK), PI3K/Akt- and JAK/STAT pathways [107]. Phase I/II trial data exist for HER2-targeted therapies in CCA. Neratinib, a pan-HER TKI, was investigated in the SUMMIT trial (phase II, NCT01953926) in patients with HER2 mutations. Patients received oral neratinib at a dosage of 240 mg daily. In the CCA cohort of 25 patients, the ORR was 16% (95% CI, 4.5–36.1). The mPFS was 2.8 months (95% CI, 1.1–3.7), and the mOS was 5.4 months (95% CI, 3.7–11.7). Although the CCA cohort did not meet the prespecified criteria for further expansion, some patients demonstrated clinically relevant tumor responses and disease control [149]. The HER2-targeted bispecific antibody zanidatamab demonstrated promising anti-tumor activity in a phase 2b trial (HERIZON-BTC-01, NCT04466891) among patients with HER2-amplified locally advanced or metastatic CCA [150]. The study enrolled 87 patients in two cohorts: cohort 1 (IHC 2+ or 3+; HER2-positive) and cohort 2 (IHC 0 or 1+). Patients received zanidatamab at a dosage of 20 mg/kg intravenously every 2 weeks. The observed ORR in cohort 1 was 41.3% (95% CI, 30.4–52.8) [150]. Remarkably, GB also benefited from targeted therapy in the HERIZON-BTC-01 study.

A study investigating the pan-HER1-4 inhibitor varlitinib in combination with gemcitabine and cisplatin as a first-line treatment for advanced or metastatic CCA (phase I/II NCT02992340) was terminated by the sponsor, despite achieving an ORR of 35% and a DCR of 87% [151]. Varlitinib was further investigated in a second-line setting in combination with capecitabine. However, in the TreeTop trial (phase II, NCT03093870), varlitinib combined with capecitabine did not demonstrate improved efficacy compared to capecitabine alone [152].

Trastuzumab stands out as a promising antibody targeting HER2. Several studies are exploring its combinations with classical cytotoxic chemotherapy (FOLFOX), antibody-drug conjugates (trastuzumab deruxtecan, trastuzumab emtansine), or in combination with other small molecules (trastuzumab plus tipifarnib), and simultaneous HER2-blockade (trastuzumab plus pertuzumab) [153,154,155,156,157]. In a phase II trial (NCT04722133), mFOLFOX6 combined with trastuzumab for second- or third-line therapy exhibited promising anti-tumor activity. The mPFS was 5.1 months (95% CI, 3.6–6.7), and the mOS was 10.7 months (95% CI, 7.9–not reached). The KAMELEON trial (phase II, NCT02999672), which investigated trastuzumab emtansine in patients with HER2-positive advanced CCA, reported recruitment difficulties, resulting in no representative data being generated from the study regarding the treatment of CCA with trastuzumab emtansine. However, a phase II trial (NCT02675829) is ongoing to evaluate anti-tumor activity with trastuzumab emtansine in HER2 overexpressing solid tumors. The combination of pertuzumab and trastuzumab was investigated in the MyPathway trial (phase II, NCT02091141). In this trial, 39 patients were enrolled and received intravenous pertuzumab (840 mg loading dose, then 420 mg every 3 weeks) plus trastuzumab (8 mg/kg loading dose, then 6 mg/kg every 3 weeks. The trial showed a promising ORR of 23% (95% CI, 11–39) [157].

The phase II basket trial SGNTUC-019 (NCT04579380) investigated the combination of tucatinib and trastuzumab for previously treated Her 2-positive CCA. In the CCA cohort, 30 patients were included with previously treated HER2 overexpressing or amplified (HER2-positive) tumors with no prior HER2-directed therapy. The ORR was 46.7% (90% CI, 30.8–63.0) and the mPFS was 5.5 months (90% CI, 3.9–8.1). The authors concluded that tucatinib combined with trastuzumab had clinically significant antitumor activity [158].

Trastuzumab in combination with gemcitabine and cisplatin in patients with treatment-naive HER2-positive CCA was investigated in the TAP trial (phase II). 876 patients were screened, of whom 90 were enrolled in the study. Notably, GB was the most frequent tumor site with 96%. Patients treated with trastuzumab plus gemcitabine/cisplatin demonstrated an mPFS of 7 months (95% CI, 6.2–7.8). A complete or partial response was observed in 55.5% of patients, while 24.4% of patients maintained stable disease as their optimal response to treatment, resulting in an overall disease control rate of 80%. The combination of gemcitabine/cisplatin and trastuzumab successfully met its primary endpoint of enhancing PFS [159]. However, additional randomized phase III studies are needed to assess whether combining trastuzumab with chemotherapy enhances survival compared to chemotherapy alone among patients with CCA and the specific group of patients with GB.

A phase 1 trial combining neratinib, trastuzumab, pertuzumab, and paclitaxel in patients with HER2 overexpression in solid tumors was terminated due to increased toxicity.

The antibody-drug conjugate trastuzumab deruxtecan (TDxd) was investigated in patients with advanced solid tumors in the DESTINY-PanTumor02 study. The phase II study (NCT04482309) evaluated T-DXd (5.4 mg/kg once every 3 weeks) for HER2-expressing locally advanced or metastatic disease after at least one prior systemic treatment or without alternative treatments. The trial enrolled 41 patients with biliary tract cancer. Among these patients, 16 showed a HER2-expressing (immunohistochemistry) status of 3+, while the other 25 showed a status of 2+. For patients with high HER2 expression (3+) the ORR was 56.3% and for patients with moderate HER2 expression (2+) the ORR was 0%. The reported mPFS was 7.4 months (95% CI, 2.8–12.5) for Her3+ tumors and 4.2 months (95% CI, 2.8–6.0) for HER2+ tumors. The reported mOS was 12.4 months (95% CI, 2.8–not reached) for Her3+ tumors and 6.0 months (95% CI, 3.7–11.7) for HER2+ tumors [160].

Based on these findings (inclusive data from DESTINY-Lung01; DESTINY-CRC02) the FDA approved TDx for unresectable or metastatic HER2-positive (IHC 3+) solid tumors, including advanced cholangiocarcinoma. Ongoing studies for molecular targeting HER2 in CCA are mentioned in Table 11. Prospective, comparative, and large-scale trials in the future will be crucial to confirm the efficacy of anti-HER2 therapies in managing CCA.

### 4.5. Target Epidermal Growth Factor Receptor (EGFR)/Human Epidermal Growth Factor Receptor (HER) 1

EGFR is a receptor tyrosine kinase that, when activated by its ligands (such as epidermal growth factor or EGF), initiates a signaling cascade involved in cell proliferation, survival, and differentiation. In CCA, alterations in the EGFR pathway, including overexpression of EGFR or its ligands, mutations in EGFR, or downstream signaling components, contribute to the development and progression of the disease. Enhanced EGFR signaling leads to uncontrolled cell growth, resistance to apoptosis, increased angiogenesis, and metastasis. EGFR overexpression or activation in CCA has been linked to poorer prognosis and aggressive tumor behavior. Therefore, targeting the EGFR pathway has been explored as a potential therapeutic strategy [7,105,161].

The clinical efficacy of EGFR inhibition in advanced or metastatic CCA was assessed in a phase II trial involving erlotinib (phase II, NCT00033462) [162]. However, in a large non-biomarker-stratified phase III trial (NCT01149122), the addition of erlotinib (a HER1/EGFR inhibitor) to gemcitabine/oxaliplatin did not demonstrate clinical benefit [163]. Erlotinib specifically targets the tyrosine kinase domain of EGFR and exhibits more efficacy against mutant EGFR than wild-type EGFR. Notably, monoclonal antibodies such as cetuximab (BINGO, NCT00552149) and panitumumab (Vecti-BIL, NCT01389414.), which selectively target wild-type EGFR, also failed to demonstrate significant anti-tumor activity in several non-biomarker-stratified phase II studies [164,165].

Given the rarity of EGFR overexpression in the pathogenesis of CCA, further targeted therapy using small molecules or monoclonal antibodies should be evaluated in subsequent biomarker-stratified studies. Additionally, the combination of EGFR and VEGF inhibition with erlotinib and bevacizumab was investigated but showed no promising clinical activity in an unselected population of patients with CCA [166].

### 4.6. Target Neurotrophic Tyrosine Receptor Kinase (NTRK)

Tropomyosin receptor kinase (TRK) are RTKs predominantly expressed in human nervous tissue. This transmembrane receptor family consists of three members: TRK A, TRK B, and TRK C [167]. Their respective encoding genes are NTRK 1, NTRK 2, and NTRK 3. The most common driver of oncogenic TRK activation is the presence of NTRK gene fusions [168]. The fusion proteins lead to constitutive and ligand-independent activation of the TRK kinase domain, initiating sustained activation of signaling pathways, such as the MAP kinase, protein kinase C (PKC), and phosphatidylinositol 3-kinase (PI3K), which ultimately stimulate proliferation or inhibit apoptosis signaling [108,169].

Three basket trials for solid tumors with NTRK fusion—the NAVIGATE trial (phase II NCT02576431) the SCOUT trial (phase I/II NCT02637687), and a safety trial (phase I NCT02122913)—enrolled and treated 55 patients with larotrectinib (100 mg twice daily). These patients represented 17 different TRK fusion-positive tumor types, including 2 patients with CCA. Across the entire collective, the ORR was 75% (95% CI, 61–85) based on independent review and 80% (95% CI, 67–90) according to investigator assessment [170]. The data led to the FDA granting accelerated approval for the use of larotrectinib in adult and pediatric patients with solid tumors harboring NTRK gene fusion. Larotrectinib has also been approved by the EMA. Notably, it should be emphasized that from the two patients with CCA in the basket studies only one patient showed a tumor response, whereas the other showed tumor progression [170].

Entrectinib is a potent inhibitor of TRK A/B/C. The pooled analysis of three phase 1 or 2 clinical trials (ALKA-372–001 NCT03066661, STARTRK-1 NCT02097810, and STARTRK-2 NCT02568267) enrolled patients with advanced or metastatic NTRK fusion-positive solid tumors and demonstrated clinically meaningful responses. Of the 54 patients who received entrectinib orally at a dose of 600 mg once daily, the median follow-up of 12.9 months revealed an ORR of 57% (95% CI, 43.2–70.8). Notably, only one patient with CCA was part of this collective. This patient showed a partial tumor response. [171]. Based on these data, both the FDA and EMA approved entrectinib for use in adult and pediatric patients with solid tumors carrying an NTRK gene fusion. However, due to the low prevalence of NTRK gene fusion in CCA (estimated by 0.75%) and the limited representation of patients with CCA in the basket trials, further studies are necessary to evaluate the clinical benefit of larotrectinib and entrectinib in advanced or metastatic CCA with NRTK gene fusion [172] (see Table 12).

### 4.7. Targeting BRAF Alterations

BRAF is a gene encoding the serine/threonine kinase B-Raf, a proto-oncogene found in mutated forms in malignant tumors. In CCA, the incidence of BRAF V600E mutations ranges from 3% to 5%, primarily occurring in iCCA [146]. As a crucial player in the EGFR-mediated MAPK pathway, BRAF impacts cell proliferation, growth, differentiation, migration, and apoptosis. Mutations in Codon 600 result in constitutive activation. Promising data targeting BRAFV600E mutations have emerged from trials such as ROAR (phase II, NCT02034110) and NCI-MATCH trial (NCT02465060) [173,174].

In the ROAR trial, 43 patients with BRAFV600E-mutated biliary tract cancer were enrolled and treated with dabrafenib (150 mg twice daily) plus trametinib (2 mg once daily), with an investigator-assessed ORR of 53% (95% CI, 37.7–68.8) and an independent reviewer-assessed ORR of 47% (95% CI, 31.2–62.3). mPFS and mOS was 9.0 months and 13.5 months, respectively [174,175]. The NCI-Match study subprotocol H included patients with solid tumors harboring BRAFV600E mutations. Patients received dabrafenib in combination with trametinib; 29 patients were included in the primary efficacy analysis and demonstrated a confirmed ORR of 38% (95% CI, 22.9–54.9). The NCI-Match study included 4 patients with CCA, all showing tumor response [173]. Based on data from these trials, and other basket trials addressing BRAFV600E-mutated tumors, the FDA approved the combination of trametinib and dabrafenib for patients with BRAFV600E-mutated solid tumors, including advanced CCA. For non-V600E BRAF-mutated CCA, the BEAVER trial (phase II, NCT03839342) is investigating the combination of binimetinib plus encorafenib [176]. Another study, evaluating the combination of dabrafenib and trametinib in Patients with BRAFV600E/R or non-V600 BRAF mutated advanced solid tumors (BELIEVE trial, phase II) reported a clinically meaningful ORR of 28%. The study enrolled 74 patients with BRAFV600E mutation (94%) and 3 patients with non-V600E mutations [177]. In the overall patient population, the mPFS was 6.5 months (95% CI, 4.2–7.2) and the reported mOS was 9.7 months (95% CI, 7.5–12.2). The trial enrolled 2 patients with GB/eCCA and 4 patients with iCCA [177]. Overall survival and progression-free survival data of the subgroups are currently pending.

Targeted molecular therapy for BRAF-mutated patients with dabrafenib and trametinib appears to be a viable therapeutic option. Ongoing clinical trials evaluating the benefit of molecular targeting in CCA are reported in Table 13.

### 4.8. Targeting Alterations in Deoxyribonucleic Acid (DNA) Damage Repair Genes (DDR Genes)

Several ongoing studies are assessing molecular-directed therapies for patients with mutations in DDR genes. Dysregulation of DDR genes is considered a critical intracellular pathway regulating proliferation, apoptosis, and chemoresistance in CCA, detected in approximately 63.5% of patients [178,179]. Specifically in iCCA, various DNA damage response-related genes exhibit mutations or alterations, including PTEN, BRCA 1/2 (1.9% and 4.4%), ATM (5.7%), ATR (5.1%), ARID1A (13%), TP53, CDKN2A, POLD1, POLE, IDH 1 (20%), BAP1 (7.4%), BARD1 (2.5%), CHK 1/2 (1% and 1.9%), PALB2 (1.9%) and PBRM1 (12%) [109,180,181]. Targeting DDR genes in cancer involves the strategy of synthetic lethality [182], wherein additional inhibition of DDR genes in tumor cells harboring mutations is employed. This approach capitalizes on inducing additional DNA damage, triggering cell death selectively in cancer cells while sparing healthy cells [183,184]. Moreover, the inhibition of DDR genes may augment the cytotoxic effects of standard cytotoxic chemotherapy used for patients with CCA. However, targeted therapy for DDR genes in CCA lacks approval from the FDA/EMA at present.

Several studies, including preclinical and clinical phase I/II trials, explore the use of poly(ADP-ribose)-Polymerase inhibitors (PARP inhibitors). PARP inhibitors function by blocking the repair of damaged DNA structures. DNA damage is a continual occurrence, and the enzyme PARP plays a crucial role in repairing breaks in the DNA chain molecule. When PARP is inhibited by PARP inhibitors, DNA fragments accumulate within the cell nucleus, activating a genetic self-destruct program known as apoptosis.

Preclinical studies have demonstrated the efficacy of PARP inhibition in cholangiocarcinoma (CCA) cell lines [185]. The PARP inhibitor niraparib was investigated in a phase II trial (NCT03207347, UF-STO-ETI-001) involving patients with BAP1 and other DDR-deficient neoplasms. In cohort A (comprising patients with mesothelioma, uveal melanoma, renal cell carcinoma, cholangiocarcinoma), 9 patients demonstrated tumor response (1 partial response, 8 stable disease), out of which 7 had a BAP1 mutation. The study did not specify the exact number of patients with cholangiocarcinoma included. However, the final results highlighted a clinical benefit from niraparib in patients with a BAP1 mutation [186]. Olaparib, another PARP inhibitor, has been studied in patients with CCA and breast cancer antigen (BRCA) mutation. A multicenter retrospective study revealed mOS estimates ranging between 40.3 months (95% CI, 6.73–108.15) and 25 months (95% CI, 15.23–40.57) for patients with CCA and BRCA mutation treated with olaparib following platinum-containing therapy [187]. Currently, a phase II trial (NCT04042831) evaluating treatment with olaparib in patients with metastatic CCA with aberrant DNA repair gene mutations is ongoing.

PARP inhibitors also hold promise for patients with CCA and IDH1/2 mutation, as the presence of these mutations enhances sensitivity to PARP inhibition [188]. Ongoing clinical trials are currently exploring PARP inhibition in patients with CCA and IDH 1/2 mutation (SOLID trial NCT03991832 or NCT03212274), as outlined in Table 14. Additionally, the investigation of combination therapies involving small molecules targeting DDR genes (such as the PARP inhibitor plus ATR inhibitor) and the combination of immunotherapy with DDR inhibition is underway, as highlighted in the following section. Combination therapy is increasingly recognized as crucial for the future, given the growing reports of resistance to PARP inhibition alone [189].

The serin/threonine kinases of the Wee1 family represent another potential target for DNA damage checkpoint inhibition [190]. Preclinical studies have demonstrated the efficacy of a combination therapy involving Wee1 inhibition alongside ATR inhibition or PARP inhibition [191,192]. A promising Wee1 inhibitor, adavosertib, is currently under investigation in clinical trials for CCA, specifically in patients with BRCA 1/2 mutation. In a phase I trial (NCT01748825) involving adavosertib administered to patients with advanced solid tumors, one patient diagnosed with CCA was enrolled and demonstrated stable disease as the best tumor response [193].

Additional promising targets for CCA treatment involve checkpoint kinases 1 and 2 (CHK1/2), pivotal in DNA replication and DNA repair. Overexpression of CHK1/2 in tumor cells augments and fortifies reliable DNA repair mechanisms. Inhibiting CHK1/CHK2 can induce DNA damage accumulation, prompting early entry into the mitotic phase of the cell cycle and subsequent tumor cell death [109]. Prexasertib, a CHK1/2 inhibitor, was investigated in a phase 1b dose-escalation study [194]. In cohort 1, part 3 (comprising prexasertib plus cisplatin), one patient diagnosed with CCA was enrolled and demonstrated a complete response lasting 3.2 months. Further clinical studies assessing CHK1/2 inhibitors are imperative to evaluate their clinical efficacy and potential benefits.

Breast cancer antigens 1/2 (BRCA1/2) are central enzymes in homologous recombination crucial for repairing double-strand breaks (DSB). In cells with BRCA1/2 mutations, the repair of DSB is inefficient, leading to the accumulation of gene alterations and carcinogenesis [195]. The treatment of tumors with BRCA1/2 mutation has been extensively studied in various solid tumors, such as breast cancer, ovarian cancer, prostate cancer, and pancreatic cancer. Numerous phase III studies demonstrate the efficacy of PARP inhibitors in BRCA1/2-mutated cancer. Furthermore, PARP inhibitors hinder the repair of single-strand breaks in the cancer cell DNA, induced by prior chemotherapy, resulting in double-strand breaks during subsequent cell division. Typically, these breaks are mended via homologous recombination. However, cancer cells with defective homologous recombination, like those with BRCA1/2 mutations, undergo apoptosis [196]. Multicenter retrospective analyses and case reports provide evidence supporting the efficacy of PARP inhibitors in CCA with BRCA1/2 mutation [187,197,198,199,200,201]. The retrospective studies as well as case reports demonstrated a PFS ranging from 2.0 to 42.6 months and OS up to 64.8 months. However, due to the limited number of cases and retrospective data analysis, there is substantial diversity in survival and tumor control data. Furthermore, case reports highlight the potential of combining PARP inhibition with immunotherapy in CCA with BRCA1/2 mutation and PD-1 overexpression, demonstrating promising anti-tumor efficacy [202,203]. Because of growing evidence for PARP inhibition in CCA with BRCA1/2 mutation there a several ongoing clinical phase I/II trials (see Table 14).

### 4.9. Targeting ROS1/ALK/MET Alterations

Anaplastic lymphoma kinase (ALK) is a receptor tyrosine kinase, and alterations in ALK are implicated in tumorigenesis. Similarly, the receptor tyrosine kinase ROS1 (prot-oncogene c-ros) is recognized as an oncogenic driver. Mutations in ALK and ROS1 are observed in approximately 3% to 9% of patients with CCA. However, data regarding the therapy for patients with CCA with ROS1 or ALK alterations is extremely limited, primarily consisting of individual case reports [204,205]. Currently, an active multicenter basket trial is underway, assessing the role of entrectinib in first or subsequent lines of treatment for patients with activated neurotrophic tyrosine receptor kinase (NTRK) 1/2/3, ALK, or ROS1 pathway alterations across various malignancies (NCT02568267).

The binding of fibroblast-derived hepatocyte growth factor (HGF) to its receptor MET induces proliferation, angiogenesis, and migration while also prompting the conversion of epithelial cells into a mesenchymal phenotype. This transition to a mesenchymal phenotype amplifies the invasiveness of malignant cells. However, in a phase II trial (NCT01954745) treating patients with advanced CCA, the unselective tyrosine kinase and MET inhibitor cabozantinib demonstrated no significant anti-tumor activity in an unselected patient population [206]. Tivantinib (ARQ 197), a selective c-MET inhibitor, underwent investigation in a phase I dose-escalation study combined with gemcitabine for patients with advanced or metastatic solid tumors (in an unselective patient population). This study enrolled 74 patients with solid tumors, including 8 patients with CCA. Among them, one patient with CCA achieved a partial tumor response [207]. In a case report involving a CCA patient with EHBP1-MET fusion, treatment with the MET inhibitor crizotinib led to a partial tumor response lasting 8 months [208]. Moving forward, further clinical trials focusing on selected patient groups with evidence of MET alterations are necessary to determine the efficacy of MET inhibition in patients with CCA and MET alterations (see Table 15).

### 4.10. Targeting CASEIN KINASE (CK2)

Casein kinase 2 (CK2) is a protein kinase involved in various cellular processes like cell proliferation, apoptosis, and DNA repair. In the context of cholangiocarcinoma (CCA), the role of CK2 in its pathogenesis is increasingly being studied, although the exact mechanisms are still being elucidated. There is evidence that CK2 is overexpressed in CCA and there is a significant association of CK2 overexpression with progression and prognosis of CCA. CK2 is known to regulate cell growth and proliferation. Dysregulated CK2 activity can contribute to uncontrolled cell division, potentially leading to tumor development and progression in CCA. Moreover, CK2 has been associated with promoting cell survival by inhibiting apoptosis (programmed cell death). In cancers, including CCA, overactive CK2 may contribute to the evasion of cell death, allowing cancer cells to persist and proliferate. CK2 can also affect pathways like Wnt/β-catenin, PI3K/AKT, and others, which are frequently dysregulated in cancers, including CCA. Nonetheless, CK2 plays a role in DNA repair processes. Dysfunctional CK2 activity might affect DNA repair mechanisms, potentially leading to genetic alterations and the accumulation of mutations [209,210].

The casein kinase 2 (CK2) inhibitor, silmitasertib, has demonstrated promising efficacy in patients with locally advanced or metastatic CCA when combined with gemcitabine/cisplatin. In a phase I/II trial, 87 patients were enrolled and received silmitasertib, with 55 patients were evaluable for efficacy. The reported results showed an mPFS of 11.1 months (95% CI, 7.6–14.7), an mOS of 17.4 months (95% CI, 13.4–25.7), and an ORR of 32.1 [211]. Confirmation of these findings through a phase III study is crucial to establish the clinical benefit definitively. Targeting CK2 or its downstream pathways could potentially offer therapeutic strategies for managing CCA. However, further studies are necessary to validate these approaches and their clinical relevance.

### 4.11. Targeting Receptor Tyrosine Kinase RET

RET rearrangements and mutations are known to be treatable drivers of carcinogenesis. RET fusion proteins and activating point mutations can trigger downstream signaling pathways, leading to uncontrolled cell proliferation, tumor progression, and oncogenesis [212]. Pralsetinib, a selective inhibitor of the RET receptor tyrosine kinase, was evaluated in the ARROW trial (phase 1/2, NCT03037385) with 29 patients harboring RET fusion-positive solid tumors, including three patients with CCA. Among these patients with CCA, one achieved stable tumor control, while the other two exhibited a partial tumor response. These findings suggest the efficacy of RET inhibition in patients with RET fusion-positive CCA [213]. Despite FDA approval of pralsetinib for treating advanced or metastatic RET-fusion-positive lung and thyroid cancers based on ARROW trial data, there is yet no approval specifically for CCA. Another highly selective inhibitor of the RET receptor tyrosine kinase is selpercatinib. In the LIBRETTO-001 trial (phase 1/2, NCT03157128), 45 patients with RET fusion-positive solid tumors (excluding lung or thyroid tumors) were enrolled, including two patients with CCA. Among all participants, 41 were evaluated for efficacy and the ORR was 43.9% (95% CI, 28.5–60.3), the mOS was 18.0 months (95% CI, 10.7–NE) and the mPFS was 13.2 months (95% CI, 7.4–26.2) [214]. Only one of the two CCA patients was evaluable for efficacy, demonstrating a tumor response lasting 5.6 months [214]. Based on the data, the FDA granted accelerated approval for selpercatinib in locally advanced or metastatic RET fusion-positive solid tumors, including CCA. Ongoing studies for molecular targeting the RET receptor tyrosine kinase shown in Table 16.

### 4.12. Targeting the PI3K/AKT/mTOR Signaling Pathway

The phosphoinositide-3-kinase (PI3K)/Akt pathway represents an important intracellular signaling pathway that regulates diverse cellular processes, including cell growth, proliferation, and metabolism. PI3K, activated by growth factors, initiates downstream signaling, culminating in the activation of the serine/threonine kinase Akt. Akt, in turn, can suppress pro-apoptotic proteins from the Bcl-2 family, thereby inhibiting apoptosis. Moreover, it stimulates protein translation through the mechanistic target of rapamycin (mTOR) and indirectly promotes cell proliferation.

Acting as a counterpart to PI3K, the phosphatase PTEN balances the activity of this pathway. Loss-of-function mutations in PTEN or activating alterations in PI3K/Akt can lead to the constitutive activation of this signaling cascade [215]. Currently, several ongoing clinical trials are underway for advanced CCA, focusing on small molecules that separately target PI3K, AKT, and mTOR (see Table 17).

A phase 2 study (NCT02631590) investigating the PI3K inhibitor copanlisib in combination with gemcitabine and cisplatin for an unselective patient population with advanced CCA revealed no significant difference in mOS when copanlisib was added to standard chemotherapy [216]. Subprotocol Z1F of the NCI-MATCH trial (phase II, NCT05490771) evaluated copanlisib, in patients with PIK3CA-mutated cancers, enrolling 35 patients, with 25 included in the primary efficacy analysis. Among the 25 patients, 6 had gastrointestinal tumors, including one CCA [217]. Although the study met its primary endpoint with a 16% ORR, the treatment response of the included CCA patients has not been reported yet. BKM120 (buparlisib), another pan-PI3K inhibitor, was investigated for efficacy in patients with malignancies harboring a PI3K pathway activation (phase II, NCT01833169). Among the 146 patients, including 6 patients with GB and 4 patients with cancer of bile ducts, clinical benefit was observed only in one patient with cancer of gall bladder ducts [218]. While buparlisib was well-tolerated overall in the study population, efficacy was limited despite the selection of aberrations in the PI3K pathway. Further studies are required to evaluate the value of buparlisib in treating tumors with PI3K pathway aberrations. MK2206, an inhibitor of serine/threonine kinase Akt, was evaluated for advanced or metastatic CCA in a phase II trial (NCT01425879). The trial enrolled 8 patients (6 iCCA, 2 eCCA) without stratification according to Alk alterations. The trial was terminated by the sponsor, and the preliminary findings suggest that MK-2206 did not demonstrate meaningful clinical activity as a single agent in patients with refractory CCA, with an mOS of 3.8 months (95% CI, 2.2–6.7) and an mPFS of 1.7 months (95% CI, 0.5–5.6) [219]. The ongoing TAPISTRY trial has enrolled patients with Akt1/2/3 mutations who will be treated with ipatasertib. Additionally, in TAPISTRY, the use of inavolisib in patients with PIK3CA mutation is being investigated.

The mTOR inhibitor everolimus was investigated in unselected patients with advanced CCA, regardless of alterations in the PI3K/AKT/mTOR signaling pathway. In the ITMO study (phase II), 39 patients with advanced CCA previously treated with chemotherapy were enrolled and received a daily oral dose of everolimus (10 mg). The mPFS was 3.2 months (95% CI, 1.8–4.0), and the mOS was 7.7 months (95% CI, 5.5–13.2). The authors concluded that everolimus demonstrated a favorable toxicity profile and promising anti-tumor activity [220]. Everolimus was further assessed in a phase II study involving patients with advanced solid tumors refractory to standard therapy harboring PIK3CA amplification/mutation and/or PTEN loss. 10 patients were enrolled, including one patient with CCA achieved tumor control [221]. Additionally, a single case report demonstrated a partial tumor response in an advanced CCA patient with a PIK3CA mutation [222]. Considering the data, the use of everolimus in patients with advanced CCA and PIK3CA alteration after the failure of established therapeutic approaches could be considered.

### 4.13. Targeting the Wnt/β-Catenin Signaling Pathway

The Wnt/β-catenin signaling pathway plays a crucial role in various cellular processes, including embryogenesis, tissue homeostasis, and carcinogenesis. Dysregulation of the Wnt/β-catenin pathway has been observed in CCA. Mutations, i.e., those of CTNNB1 and AXIN1, or alterations in components of the Wnt/β-catenin pathway, can lead to its abnormal activation. This activation is associated with uncontrolled cell proliferation, inhibition of apoptosis, and increased tumor invasiveness [223,224,225]. Several preclinical in vitro studies have demonstrated the role of Wnt/β-catenin signaling in inducing malignancy in CCA cells. [226]. Another target, influencing the Wnt/β-catenin signaling pathway is Dickkopf 1 (DKK1). DKK1 is a secreted protein that plays a pivotal role in modulating the Wnt/β-catenin signaling pathway. DKK1 is known for its ability to antagonize the Wnt/β-catenin signaling pathway. It achieves this by binding to the Wnt co-receptor LRP5/6, thus preventing Wnt ligands from binding and thus blocking the activation of downstream signaling pathways. Studies have shown that DKK1 expression is elevated in cholangiocarcinoma tissues compared to normal bile duct tissues. This upregulation suggests a potential role for DKK1 in promoting CCA development. Despite its role as a Wnt pathway antagonist, in certain contexts, elevated levels of DKK1 in CCA have been associated with tumor progression [227,228]. However, the role of targeting WNT/β-catenin signaling requires further investigation and validation through preclinical and clinical studies (see Table 18).

### 4.14. Targeting Cyclin Dependent Kinase 4/6 (CDK4/6)

In many cancers, including CCA, overexpression or dysregulation of CDK4/6 leads to increased cell proliferation. Mutations or amplifications of genes encoding these proteins can disrupt the normal cell cycle control mechanisms, contributing to tumor growth. Drugs like palbociclib, ribociclib, and abemaciclib are examples of CDK4/6 inhibitors approved for certain types of breast cancer. The effects of CDK4/6 inhibitors on cholangiocarcinoma (CCA) cell lines have indeed been explored in a limited number of studies, and the results have been somewhat conflicting or inconclusive. While some studies have shown potential therapeutic effects, others have reported limited efficacy or conflicting outcomes. Clinical trials are ongoing to assess their efficacy in various other cancers, including CCA (see Table 19) [229].

### 4.15. Targeting the RAS/RAF/MEK/ERK Signaling Pathway

The RAS/RAF/MEK/ERK pathway stands as a pivotal signaling pathway governing cell growth, cell proliferation, and cell motility. Typically initiated by signals at the epidermal growth factor receptor (EGFR), a receptor tyrosine kinase (RTK), the pathway holds exceptional importance in human tumors. Constitutive activation of EGFR drives growth-promoting effects in tumor cells, with RAS protein mutations, in particular, leading to sustained, receptor-independent activation of downstream RAF proteins [230]. KRAS, a downstream player in this pathway, also influences the PI3K/AKT/mTOR pathway. The incidence of KRAS mutations in CCA displays significant variability across study populations and tumor locations. Initially deemed undruggable, KRAS mutations have seen a recent breakthrough with the development of drugs targeting KRAS^G12C^ mutations. However, the prevalence of the KRASG12C mutation in CCA remains notably low. A study examining circulating tumor DNA in CCA patients revealed a mere 1.2% frequency of the KRASG12C mutation [231].

The KRYSTAL-1 trial (phase I/II, NCT03785249), assessed adagrasib, a selective and irreversible KRAS^G12C^ inhibitor, in patients with advanced solid tumors carrying the KRAS^G12C^ mutation. This trial enrolled 64 patients, including 12 patients with advanced CCA. Among the 57 patients with measurable disease, the ORR was 35.1%. The mPFS was 7.4 months (95% CI, 5.3–8.6), and the mOS was 14.0 months (95% CI, 8.5–18.6). Specifically focusing on the 12 patients with CCA, the ORR was 41.7%, with an mPFS of 8.6 months (95% CI, 2.7–11.3) and an mOS of 15.1 months (95% CI, 8.6–not estimable) [232]. Based on these results, the authors concluded that adagrasib demonstrates promising clinical activity in pretreated patients with CCA. Yet, adagrasib does not currently have FDA/EMA approval for use in KRASG12C mutant CCA.

Another specific KRAS^G12C^ inhibitor, sotorasib, has been recently approved by the FDA for the therapy of previously treated non-small-cell lung cancer with KRAS^G12C^ mutation. In the CodeBreaK 100 trial (NCT03600883), a basket trial for KRAS^G12C^ mutated solid tumors, sotorasib demonstrated promising anti-tumor effects in a subgroup of patients with advanced pancreatic cancer and KRAS^G12C^ mutation [233]. However, only one patient with CCA was included in the CodeBreaK 100 trial, making a definitive assessment of sotorasib’s benefit in KRASG12C mutated CCA inconclusive. The investigation of KRAS inhibitors has gained significant momentum recently, especially due to the positive data from existing compounds. Currently, numerous KRAS^G12C^- and pan-KRAS inhibitors are in various stages of development and testing (see Table 20).

Targeted therapy stands as a potential avenue for inhibiting the RAS/RAF/MEK/ERK signaling pathway by modulating downstream mechanisms within the KRAS pathway. Selumetinib, a MEK1/2 inhibitor, underwent evaluation in a phase II trial (NCT00553332) involving patients with advanced CCA. Out of 28 enrolled, 25 had tissue samples evaluated for KRAS mutations, revealing two positive cases (G12S and G12D). The authors concluded a potential efficacy of targeting MEK with selumetinib in the overall population [234]. Additional clinical trials are required to reinforce this conclusion. The ABC-04 trial (phase I, NCT01242605, dose-finding) evaluated the combination of selumetinib with cisplatin and gemcitabine in advanced or metastatic CCA. A subsequent phase II trial (NCT02151084) did not demonstrate the added benefit of selumetinib plus gemcitabine/cisplatin over gemcitabine/cisplatin alone [235]. Currently, there are no randomized phase II or III trials specifically addressing the efficacy of gemcitabine/cisplatin and selumetinib with consideration of specific molecular profiles.

The efficacy of trametinib in patients with advanced CCA was investigated in the SWOG S1310 trial (phase II, NCT02042443), where 44 unselected patients were randomized to Arm A: trametinib or Arm B: 5-FU intravenous or capecitabine orally. The mOS was 4.3 months for arm A and 6.6 months for arm B, while the mPFS was 1.4 months for arm A and 3.3 months for arm B. These results suggest no benefit for trametinib over 5-FU/capecitabine in a patient population not stratified by alterations in the RAS/RAF/MEK/ERK signaling pathway.

The combination of pazopanib (an oral VEGF receptor tyrosine kinase inhibitor) with trametinib demonstrated initial clinical efficacy in a phase I study (NCT01438554) but failed to meet the primary endpoint [236]. Conversely, a phase I study (NCT02773459) exploring binimetinib (MEK1/2 inhibitor) and capecitabine in patients with advanced or metastatic CCA with RAS/RAF/MEK/ERK pathway alterations showed encouraging anti-tumor effects. Among 34 enrolled patients, those with RAS/RAF/MEK/ERK pathway mutations demonstrated significantly greater benefits regarding PFS (5.4 vs. 3.5 months, *p* = 0.010) and OS (10.8 vs. 5.9 months, *p* = 0.160) compared to patients with wild-type status [237]. In conclusion, for patients with RAS/RAF/MEK/ERK pathway alterations and failed standard therapies, a therapeutic approach involving capecitabine and binimetinib could be considered within the framework of an individualized therapeutic approach.

### 4.16. Targeting Vascular Endothelial Growth Factor (VEGF) Signaling Pathway

VEGFs constitute a protein family involved in various physiological processes, including angioneogenesis and lymphangiogenesis. All VEGF family members elicit a cellular response by binding to the VEGF receptor (VEGFR), a tyrosine kinase that facilitates the transmission of extracellular signals into the cell interior. within this family, three receptors are identified as VEGFR 1-3. Notably, increased VEGF expression is observed in numerous tumors, contributing to enhanced blood vessel permeability. This effect disrupts the blood–tumor barrier, leading to tumor growth and increased potential for metastasis [161,238].

The monoclonal antibody (mAb) bevacizumab has been investigated in several phase II trials for the treatment of CCA. In the phase II trial (NCT00361231), 35 patients with advanced or metastatic CCA received bevacizumab in combination with gemcitabine plus oxaliplatin in the first-line setting. The mPFS was 7.0 months (95% CI, 5.3–10.3), with a 6-month PFS was 63%. However, the study was deemed formally negative as the primary endpoint, achieving a 6-month PFS of 70%, was not met [239]. Subsequent retrospective analysis demonstrated an advantage for the combination of bevacizumab plus gemcitabine/oxaliplatin over gemcitabine/oxaliplatin. In this analysis involving 57 patients (bevacizumab plus gemcitabine/oxaliplatin *n* = 32 and gemcitabine/oxaliplatin *n* = 25), the mPFS was 6.48 months for the bevacizumab group compared to 3.72 months for the gemcitabine/oxaliplatin group. Moreover, the mOS was 11.31 months with bevacizumab plus gemcitabine/oxaliplatin and 10.34 months with gemcitabine/oxaliplatin [240]. Furthermore, a phase II trial assessed the combination of FOLFIRI plus bevacizumab as a second-line therapy for metastatic iCCA following the failure of gemcitabine/oxaliplatin. This trial enrolled 13 patients, achieving an ORR of 38.4% (95%, CI 12.5–89), with an mPFS of 8 months (95% CI, 7–16) and an mOS of 20 months (95% CI, 8–48) [241]. Yet, to conclusively evaluate the value of adding bevacizumab to cytotoxic chemotherapy like gemcitabine/oxaliplatin or FOLFIRI, further phase III studies with prospective designs are necessary. Contrarily, the addition of bevacizumab to gemcitabine/capecitabine did not yield improved outcomes in an unselected group of patients with advanced CCA (NCT01007552) [242].

Another phase II trial investigated the combination of bevacizumab and erlotinib in 49 patients with advanced or metastatic CCA, resulting in an mOS of 9.9 months. This combination demonstrated clinical activity, with six patients showing partial response and 25 patients achieving stable disease [243]. Notably, patient inclusion was irrespective of mutations in the EGFR pathway. However, subgroup analysis revealed no benefit of therapy with bevacizumab and erlotinib for patients with KRAS mutations. Future studies should evaluate this combination in a molecularly gate-selected collective.

A phase II trial (NCT01206049) evaluated the addition of bevacizumab to gemcitabine/oxaliplatin compared to the combination of panitumumab plus gemcitabine/oxaliplatin (for KRAS wild-type patients). Unfortunately, the study did not meet its primary endpoint of progression-free survival. Given the higher response rate observed in the panitumumab arm, there is a possibility of considering an assessment of this combination in the neoadjuvant setting [244]. Studies for patients with advanced CCA evaluating the use of bevacizumab or biosimilars are shown in Table 21.

The mAbs aflibercept (targeting VEGFR 1/2) and ramucirumab (targeting VEGFR 2) are currently being investigated in the context of advanced or metastatic CCA. In a phase II study evaluating ramucirumab in individuals with advanced or metastatic CCA who had previously received gemcitabine-based chemotherapy, 61 patients were enrolled. The trial reported an mPFS of 3.2 months (95% CI, 2.1–4.8) and an mOS of 9.5 months (95% CI, 5.8–13.6). These findings align with known survival data in second-line therapy, suggesting considerable antitumor activity. The data were similar to known survival data in second-line therapy, suggesting appropriate antitumor activity [245].

However, in a basket trial investigating ramucirumab in combination with pembrolizumab for advanced solid tumors (NCT02443324), no patients with CCA were included among the 93 enrolled individuals. Consequently, the impact of the ramucirumab and pembrolizumab combination in patients with advanced CCA remains uncertain [246]. Another trial (phase II, NCT02711553) investigating the addition of ramucirumab to gemcitabine/cisplatin in patients with molecularly unselected, locally advanced, or metastatic CCA did not demonstrate improved PFS [247]. Similarly, the efficacy of aflibercept in combination with capecitabine was examined in the MOMENTUM trial (phase I, NCT01843725) involving patients with advanced metastatic breast and digestive cancer. Unfortunately, no patients with CCA could be included in this study, leaving the role of aflibercept in CCA treatment unclear [248].

Sorafenib, an oral multi-kinase inhibitor, targets VEGF and inhibits the serine/threonine kinase Raf, thus disrupting the Ras/Raf/MAPK signaling pathway. Although extensively studied in the treatment of HCC, multiple phase II trials have explored its potential for treating patients with advanced CCA. In a phase II trial involving 46 patients with advanced CCA, sorafenib demonstrated a DCR of 32.6% at 12 weeks, an mPFS of 2.3 months and an mOS of 4.4 months. As a single agent, sorafenib showed only modest anti-tumor activity based on survival data [249]. The SWOG 0514 trial, a phase II study of sorafenib in patients with unresectable or metastatic CCA, was closed due to the absence of ORR in patients treated with sorafenib [250]. Similarly, sorafenib plus capecitabine failed to show a clinically meaningful benefit compared to capecitabine alone, in a phase II trial including 102 patients with advanced or metastatic CCA [251]. Also, the combination of sorafenib and gemcitabine/cisplatin did not improve efficacy in a phase II trial (NCT00919061) with a reported mPFS of 6.5 months (95% CI, 3.5–8.3) and an mOS of 14.4 months (95% CI, 11.6–19.2), versus known survival data for standard gemcitabine/cisplatin therapy (ABC-02 trial for gemcitabine/cisplatin in the first-line setting: mPFS 8.0 months, mOS 11.7 months) [252]. Sorafenib in combination with gemcitabine/oxaliplatin (NCT00955721) and gemcitabine/capecitabine (NCT00634751) were part of phase II clinical trials, but conclusive data are lacking at present. In summary, the use of sorafenib, either alone or in combination with classical chemotherapy, cannot be currently recommended. Although some individual studies indicate potential anti-tumor effects, further validation through large multicenter studies is necessary.

Of note, lenvatinib, targeting VEGFR1/2/3, PDGFR, FGFR, KIT, and RET, exhibited promising anti-tumor activity in a phase I clinical trial (LENABC, NCT04656249) involving previously treated patients with CCA. The study enrolled 41 patients, with an ORR of 12% (95% CI, 1.7–22.7), an mPFS of 3.8 months (95% CI, 1.3–6.3), and an mOS of 11.4 months (95% CI, 6.6–16.2) [253]. However, in a phase II trial investigating the addition of lenvatinib to gemcitabine/oxaliplatin chemotherapy for patients with advanced iCCA, only modest efficacy was observed [254]. Additionally, a phase I trial (NCT03895970) investigated the combination of lenvatinib and immunotherapy in patients with advanced CCA. Involving 32 patients, the trial reported an ORR of 25%, an mPFS of 4.9 months (95% CI, 4.7–5.2 months) and an mOS of 11.0 months (95% CI, 9.6–12.3 months) [255]. In addition to these findings, results of the phase 2 LEAP-005 trial (NCT03797326) suggest promising anti-tumor activity when combining lenvatinib with pembrolizumab in previously treated patients with advanced CCA. In this basket trial, involving 31 patients with CCA, the observed ORR was 10% and the mPFS was 6.1 months with an mOS of 8.6 months [256]. These data indicate encouraging efficacy of lenvatinib and pembrolizumab beyond first-line therapy in advanced CCA, prompting an expansion of the cohort to include 100 patients. Lenvatinib is among the best-tolerated TKIs. Therefore, based on the presented data, there are currently several promising phase II and phase III trials in clinical research (see Table 22).

Regorafinib is another oral multi-kinase inhibitor targeting VEGFR, TIE2, KIT, RET, PDGFR, and FGFR [257]. Regorafenib demonstrated promising anti-tumor efficacy in various phase II trials involving patients with advanced metastatic CCA. In a phase II trial (NCT02053376) involving chemotherapy-refractory patients with advanced metastatic CCA, treatment with regorafenib as a single-agent therapy showed notable efficacy. Among 45 enrolled patients, 34 patients were evaluable for analysis of tumor response. The study reported an mPFS of 15.6 weeks (90% CI, 12.9–24.7) and an mOS of 31.8 weeks (90% CI, 13.3–74.3). Notably, 11% of patients achieved partial response (*n* = 5) and 44% showed stable disease (*n* = 19) [258]. In another phase II trial (NCT02115542) regorafenib as a single regimen exhibited an mPFS of 2.8 months (95% CI, 1.1–4.5) and an mOS of 7.9 months (95% CI, 0–18.7) [259]. The REACHIN trial (phase II, NCT02162914) further investigated the efficacy of regorafenib in patients with advanced CCA after the failure of gemcitabine and platinum-based chemotherapy. In this trial, 66 patients were enrolled and randomized to receive regorafenib or placebo. The regorafenib group demonstrated an mPFS of 3.0 months (95% CI, 2.3–4.9) and an mOS of 5.3 months (95% CI, 2.7–10.5). In contrast, the placebo group showed an mPFS of 1.5 months (95% CI: 1.2–2.0) and an mOS of 5.1 months (95% CI, 3.0–6.4). Overall, regorafenib resulted in a statistically significant improvement in mPFS compared to the placebo in second- and third-line treatments for pretreated advanced CCA [260]. Ongoing studies for regorafenib are mentioned in Table 23.

Vandetanib is a kinase inhibitor primarily targeting VEGFR2, EGFR, and RET [261]. Vandetanib was studied in combination with gemcitabine/cisplatin in patients with metastatic CCA in several trials. Initially evaluated in a phase I study involving 23 patients, a daily oral dose of 300 mg vandetanib was well-tolerated, paving the way for subsequent phase II trials [262]. The VanGogh trial (phase II, NCT00753675) evaluated the efficacy of vandetanib alone against the combination of vandetanib plus gemcitabine or gemcitabine plus placebo in patients with advanced or metastatic CCA. A total of 173 patients were enrolled, and the primary endpoint was PFS. The trial reported mPFS of 105 days (95% CI, 72–155) for the vandetanib group, 114 days (95% CI, 91–193) for the combination of gemcitabine plus vandetanib and 148 days (95% CI, 71–225) for gemcitabine plus placebo, showing no statistical difference. The authors concluded that vandetanib treatment did not improve PFS [263].

Cediranib, an oral pan-VEGFR inhibitor, was studied in the ABC-03 trial (phase II, NCT00939848), which enrolled 124 previously untreated patients with advanced CCA. The patients were randomly assigned to two groups: Group A received gemcitabine/cisplatin plus cediranib, while Group B received gemcitabine/cisplatin plus placebo. The trial reported an mPFS of 8.0 months (95% CI, 6.5–9.3) in the cediranib group and 7.4 months (95% CI, 5.7–8.5) in the placebo group. Notably, there was no statistically significant difference in mPFS between the two groups (HR 0.93; *p* = 0.72) [264].

In a phase II trial (SUN-CK, NCT01718327), the multi-kinase inhibitor sunitinib (targeting VEGFR1/2/3, PDGFR, C-Kit) was assessed in patients with advanced CCA who received one prior line of palliative chemotherapy. The recommended dose of sunitinib was 37.5 mg orally daily, and 53 enrolled patients received sunitinib as monotherapy. The trial reported an mOS of 9.6 months (95% CI, 5.8–13.1) and an mPFS of 5.2 months [265]. The ABC-06 study investigated FOLFOX as second-line therapy in patients with CCA, revealing an mOS of 6.2 months. When compared to this established therapy, sunitinib demonstrated promising tumor control as an oral monotherapy with an mOS of 9.6 months in the SUN-CK trial. Despite these findings, sunitinib currently lacks approval for the treatment of CCA.

Axitinib, an orally pan-VEGFR inhibitor, was evaluated in a small cohort of 5 patients with advanced and metastatic CCA following the failure of first-line gemcitabine-based chemotherapy. Axitinib was administered at a dose of 5 mg twice daily, demonstrating manageable side effects. PFS ranged from 2.0 to 19.9 months and OS from 1.5 to 7.4 months [266]. In a multicenter phase II trial involving 19 patients with advanced CCA refractory to gemcitabine-based chemotherapy, axitinib exhibited only modest activity. The trial reported an mPFS of 2.8 months (95% CI, 2.1–4.1) and an mOS of 5.8 months (95% CI, 3.3–9.7). Further clinical phase II/III studies are warranted to comprehensively explore the potential benefit of axitinib.

Surufatinib, a novel small-molecule inhibitor, exhibits a unique dual action by targeting both tumor angiogenesis (VEGFR1/2/3 and FGFR1) and immune evasion through the macrophage colony-stimulating factor 1 (CSF1) receptor. In a multicenter phase II trial, 39 patients with advanced CCA received surufatinib as second-line therapy at a dose of 300 mg once daily. The trial reported a median progression-free survival (mPFS) of 3.7 months and a median overall survival (mOS) of 6.9 months [267]. Further analysis within subgroups revealed that patients with elevated serum markers of CEA and CA19-9 had a poorer outcome. Moreover, individuals with iCCA experienced more substantial benefits from surufatinib monotherapy compared to patients with eCCA or GB [267].

Anlotinib, another novel orally active tyrosine kinase inhibitor targeting VEGFR, FGFR, PDGFR, and c-kit, was assessed in combination with sintilimab, a PD-1 inhibitor, in a phase II trial involving patients with unresectable iCCA. In this trial, 18 patients received sintilimab (200 mg, intravenously, day 1) and anlotinib (12 mg, orally, days 1–14) every three weeks, with the primary endpoint being ORR. The trial reported an ORR of 33.3%, including three patients with a complete response and three patients with a partial response. The mPFS was 7.49 months (95% CI, 3.12–13.2), while the mOS has not yet been reported [268]. Additionally, anlotinib combined with TQB2450 (an anti-PD-L1 mAb) showed promising efficacy in another phase I trial involving patients with advanced CCA who had progressed, declined, or were ineligible for first-line chemotherapy. This trial reported an ORR of 21.21%, an mPFS of 6.24 months (95% CI, 4.11–8.25), and an mOS of 15.77 months (95% CI, 10.74–19.71) [269]. Considering the objective responders observed in both trials, further clinical investigations are warranted to explore the combination of anlotinib and immune checkpoint blockade (see Table 24).

### 4.17. Immune Checkpoint Inhibitor (CPI)

Immune checkpoint inhibitors (CPI) have recently emerged as treatment options for different gastrointestinal cancers. Several clinical trials have explored the potential use of various checkpoint inhibitors in treating CCA.

Pembrolizumab monotherapy gained FDA approval for advanced CCA treatment following the KEYNOTE-158 trial (NCT02628067) in patients who had previously undergone at least one systemic therapy. Eligibility required laboratory evidence of microsatellite instability-high (MSI-high) or mismatch repair deficiency (dMMR) in their tumors.

In the KEYNOTE-158 trial, among patients with MSI-high tumors, the ORR was 30.8% (95% CI, 25.8–36.2) [270]. The mPFS was 3.5 months (95% CI, 2.3–4.2), and the mOS was 20.1 months (95% CI, 14.1–27.1) [271]. A separate phase II study (NCT01876511) involving pembrolizumab and patients with microsatellite unstable (MSI) tumors reported an ORR of 40% [272]. In the CCA-specific subset KEYNOTE-158, which enrolled 22 patients, two patients achieved a complete response, and 7 showed a partial response, amounting to an ORR of 40.9%. The mPFS for this cohort was 4.2 months (95% CI, 2.1–not reached), and the mOS was 24.3 months (95% CI, 6.5–not reached) [271]. Approximately 10% of iCCA, 5–13% of eCCA, and 5% of GB cases demonstrate MSI-high status [273]. Notably, a retrospective analysis of KEYNOTE-158 demonstrated that pembrolizumab is beneficial for tumors with a high mutational burden (TMB-H). The FDA approved pembrolizumab for TMB-H solid tumors (TMB > 10 mutations per megabase). It should be emphasized that no patient with CCA in KEYNOTE-158 exhibited TMB-high status. TMB distribution varies significantly by CCA location, with eCCA (18%) and GB (22%) showing higher rates compared with iCCA (13%) [274]. Presently, there is no established TMB-high threshold. Therefore, the reliability of previous studies is limited due to differing TMB-high definitions. Further investigations are imperative to determine a TMB cut-off beneficial for immunotherapy in solid tumors, including CCA.

Despite investigations into CPI utilization in CCA irrespective of MSI status, outcomes varied. In a phase Ib trial (KEYNOTE-028, NCT02054806), the efficacy of pembrolizumab was investigated in 20 cohorts with diverse advanced solid tumors displaying positive programmed death ligand 1 (PD-L1) status. within the CCA cohort, 24 patients were enrolled, with PD-L1 positivity defined as >1%. Results from KEYNOTE-028 revealed an ORR of 13.0% (95% CI, 2.8–33.6). The mPFS was 1.8 months (95% CI, 1.4–3.1), while the mOS was 5.7 months (95% CI, 3.1–9.8) [275]. In a subsequent phase II trial (KEYNOT-158, NCT02628067), pembrolizumab treatment was administered in patients with advanced solid tumors without considering PD-L1 status, encompassing 104 patients in the CCA cohort. In this trial, the ORR was 5.8% (95% CI, 2.1–12.1), with an mPFS of 2.0 months (95% CI, 1.9–2.1) and an mOS of 7.4 months (95% CI, 5.5–9.6). Notably, in KEYNOTE-158, all patients with CCA showing a tumor response had non-MSI-high status. Subgroup analysis based on PD-L1 expression demonstrated an ORR of 6.6% (95% CI, 1.8–15.9) and an mPFS of 1.9 months in patients with PD-L1-positive tumors Conversely, patients with PD-L1 negative tumors exhibited an ORR of 2.9% (95% CI, 0.1–15.3) and an mPFS of 2.1 months [275]. A prospective cohort study (NCT03695952) involving 39 enrolled Korean patients reported an 11.1% ORR for pembrolizumab in advanced PD-L1-positive CCA (PD-L1 > 1%) [276].

The efficacy and safety of nivolumab for metastatic CCA were investigated in a phase I trial that enrolled 30 patients. The observed ORR was 20%, and the mPFS was 3.1 months (95% CI, 2.13–4.06) [277]. Another phase I trial investigated nivolumab in patients with advanced CCA, either as monotherapy or in combination with gemcitabine/cisplatin. In the nivolumab monotherapy cohort (cohort A) comprising 30 patients who had failed previous gemcitabine-based chemotherapy, the mOS was 5.2 months (90% CI, 4.5–8.7), and the mPFS was 1.4 months (90% CI, 1.4–1.4). In cohort B, which included 30 therapy-naive patients receiving combination therapy, the mOS was 15.4 months (90% CI, 11.8–not estimable), and the mPFS was 4.2 months (90% CI, 2.8–5.6). Tumor response significantly varied between the cohorts; while only one patient in cohort A showed an objective response, compared to 11 of 30 patients in cohort B demonstrating an objective response, resulting in an ORR of 13% [278]. In a phase II study (NCT02829918) investigating nivolumab monotherapy in pretreated patients with advanced CCA, 54 patients were enrolled. The mPFS was 3.68 months (95% CI, 2.30–5.69 months), and the mOS was 14.24 months (95% CI, 5.98–not reached). The investigator-assessed objective response rate was 22% [279].

Th PD-1 mAb durvalumab was investigated in an Asian population with advanced CCA, esophageal, or head-and-neck cancer. The phase I trial (NCT01938612) enrolled 107 patients, including 42 patients with CCA, who received durvalumab in the monotherapy cohort. In the CCA durvalumab cohort, the mOS was 8.1 months (95% CI, 5.6–10.1), and the ORR was 4.8% [280,281]. In unselected patient collectives, current study data regarding the use of CPI monotherapy indicate, at best, moderate anti-tumor activity. Ongoing clinical research is focusing on further phase II studies and exploring combinations of immunotherapeutic agents with classical chemotherapy. The efficacy of combining the PD-L1 inhibitor durvalumab with the CTLA-4 inhibitor tremelimumab was assessed in a phase I trial (NCT01938612) involving an Asian population with advanced CCA. In the durvalumab plus tremelimumab cohort (overall 124 patients, 65 patients with CCA), the mOS was 10.1 (95% CI, 6.2–11.4), and the ORR was 10.8% [280,281].

The combination of durvalumab and tremelimumab with gemcitabine/cisplatin underwent investigation in a phase II trial (NCT03046862). The trial enrolled 128 patients across three cohorts: cohort A (32 patients) received chemotherapy followed by chemotherapy plus durvalumab and tremelimumab; cohort B (49 patients) received chemotherapy plus durvalumab; cohort C (47 patients) received chemotherapy plus durvalumab and tremelimumab. Overall, 82 of 124 patients (66%) were evaluated for tumor response, showing an ORR of 50% in cohort A, 72% in cohort B, and 70% in cohort C [91]. The mPFS was 13.0 months (95% CI: 10.1–15.9) in cohort A, 11.0 months (95% CI, 7.0–15.0) in cohort B and 11.9 months (95% CI, 10.1–13.7) in cohort C. The mOS was 15.0 months (95% CI, 10.7–19.3) in cohort A, 18.1 months (95% CI, 11.3–24.9) in cohort B, and 20.7 months (95% CI, 13.8–27.6) in cohort C. Initial therapy combining classical chemotherapy with CPI (cohort B/C) demonstrated superiority over initial chemotherapy alone (cohort A). However, the double checkpoint blockade (cohort B vs. cohort C) did not provide additional clinical benefit. Subsequently, based on these findings, the combination of durvalumab and gemcitabine/cisplatin underwent investigation in the phase III trial TOPAZ-1 (NCT03875235).

In TOPAZ-1, 685 treatment-naive patients were enrolled and randomly assigned to receive durvalumab or placebo alongside gemcitabine/cisplatin. The hazard ratio for overall survival was 0.80 (95% CI, 0.66–0.97; *p* = 0.021). The ORR was 26.7% in the durvalumab group and 18.7% in the placebo group. The mOS was 12.8 months (95% CI, 11.1–14) in the durvalumab group and 11.5 months (95% CI, 10.1–12.5) in the placebo arm (hazard ratio 0.80; 95% CI, 0.66–0.97, *p* = 0.021). Additionally, the mPFS was 7.2 months (95% CI, 6.7–7.4) in the durvalumab group and 5.7 months (95% CI, 5.6–6.7) in the placebo group [282].

Following the results of the TOPAZ-1 trial, gemcitabine/cisplatin plus durvalumab received approval from the FDA and EMA and is now recommended in international guidelines as the new standard of care for the first-line treatment of locally advanced or metastatic CCA, irrespective of PD1/PD-L1 or MSI status. In the subgroup analysis, there was hardly any benefit for patients with GB with regard to overall survival (HR 0.94). eCCA and iCCA benefited in the same way regarding overall survival (HR 0.76) [91]. In a subsequent phase II trial, IMMUCHEC (NCT03473574), the efficacy of tremelimumab in combination with gemcitabine/cisplatin plus durvalumab was reevaluated. Ultimately, the trial did not indicate a clear clinical benefit from the addition of tremelimumab [283].

Another phase III trial (KEYNOTE-966, NCT04003636) investigated pembrolizumab in combination with gemcitabine/cisplatin compared to gemcitabine/cisplatin alone for patients with advanced CCA. A total of 1069 patients were enrolled, 533 patients were randomly assigned to receive pembrolizumab plus gemcitabine/cisplatin, and 536 were randomly assigned to receive a placebo plus gemcitabine/cisplatin. The reported mOS was 12.7 months (95% CI, 11.5–13.6) in the pembrolizumab plus gemcitabine/cisplatin group compared to 10.9 months (95% CI, 9.9–11.6) in the placebo group [93]. The FDA and EMA approved pembrolizumab in combination with gemcitabine/cisplatin as first-line therapy for locally advanced or metastatic CCA.

Notably, in the KEYNOTE-966 study, PD-L1 status was also not shown to be a predictive molecular marker. In the KEYNOTE-966 subgroup analysis, only the iCCA benefited with clinically relevant improvement in the outcome with regard to overall survival (HR 0.76). GB with an HR of 0.96 and eCCA with an HR of 0.99 have hardly any benefit from the addition of pembrolizumab [93]. Additionally, the combination of pembrolizumab and chemotherapy was investigated in the second-line setting. In a phase II trial (NCT03111732), 11 patients with advanced CCA, including 8 who had failed at least one prior treatment line, were enrolled to receive pembrolizumab plus capecitabine/oxaliplatin. The mPFS was 4.1 months, with three patients achieving a partial response and six patients showing stable disease [284].

The phase II BilT-01 trial (NCT03101566) aimed to assess the combination of nivolumab plus ipilimumab against gemcitabine/cisplatin plus nivolumab in the first-line setting for patients with advanced CCA. In the gemcitabine/cisplatin plus nivolumab arm, the mPFS and mOS were 6.6 months and 10.6 months, respectively, while in the nivolumab plus ipilimumab arm, the corresponding figures were 3.9 months and 8.2 months. The primary study endpoint was achieving an 80% 6-month PFS. However, the observed 6-month PFS was 59.4% (95% CI, 40.5–74.0) in the gemcitabine/cisplatin plus nivolumab arm and 21.2% in the nivolumab/ipilimumab group. Regrettably, the phase II study failed to meet its primary endpoint [285].

The efficacy of the combination of camrelizumab plus gemcitabine/oxaliplatin as the first-line treatment for patients with advanced CCA was evaluated in a phase II trial (NCT03486678). A total of 39 patients were enrolled, and the primary endpoint was a 6-month PFS rate. The trial observed a 6-month PFS rate of 50% (95% CI, 33–65), with an mPFS of 6.1 months and an mOS of 11.8 months [286]. The study successfully met its primary endpoint; however, it is important to note that the null hypothesis was set relatively low, assuming a 6-month PFS rate of 40%. Currently, a phase II trial (NCT03796429) is investigating toripalimab in combination with chemotherapy (gemcitabine/S-1) as the first-line treatment for patients with advanced CCA. Preliminary results after enrolling 48 patients indicate an ORR of 27.1%. The mPFS was 7.0 months (95% CI, 5.5–9.1), and the mOS was 16.0 months (95% CI, 12.1–unreachable) [287].

In the phase II trial, IMbrave 151 (NCT04677504) the first-line setting for patients with advanced CCA was investigated by comparing the combination of atezolizumab/bevacizumab with gemcitabine/cisplatin (arm A) against the combination of atezolizumab with gemcitabine/cisplatin (arm B) involving 162 randomized patients. The mPFS was 8.4 months for arm A and 7.9 months for arm B. The ORR was 24.1% for arm A (95% CI, 15.1–35.0) and 25.3% for arm B (95% CI, 16.4–36.0) [288]. A publication that is under preparation and includes subgroup analysis aims to identify which patients benefit the most.

Another phase II study (NCT03951597) evaluated concurrent PD-1/VEGF blockade plus chemotherapy as the first-line treatment for advanced CCA. In this trial, 30 patients were enrolled and received toripalimab, lenvatinib, and gemcitabine plus oxaliplatin. Initial data analysis revealed impressive tumor response rates, with an ORR of 80%. Among them, 23 patients achieved a partial response, and one patient showed a complete tumor response. The mOS was 22.5 months, while the mPFS was 10.2 months. Notably, 21 of 30 patients had DNA damage response (DDR)-related gene mutations [289].

In summary, compelling clinical evidence currently supports the use of a combination of chemotherapy and immunotherapy as the first-line therapy for patients with advanced CCA. However, the extent to which chemotherapy-free protocols utilizing single immune checkpoint blockade or dual immune checkpoint blockade provide value requires further clarification through additional clinical trials (see Table 25, Table 26 and Table 27).

Bintrafusp alfa is a novel bifunctional fusion protein targeting PD-L1 and transforming growth factor (TGF beta). While data for the use of bintrafusp alfa in patients with advanced CCA are limited, promising data have emerged from phase I and phase II trials.

In a phase I trial (NCT02699515) involving 30 patients with advanced and previously treated CCA, treatment with bintrafusp alfa (1200 mg, IV, administered every 2 weeks) demonstrated a manageable safety profile [290]. The mOS was 12.7 months (95% CI, 6.7–15.8), with a 12-month OS rate was 52.0%. The ORR was 23.3% (7 responders). These promising results led to the initiation of a phase II trial (NCT03833661) for second-line treatment and a phase III trial (pending) for first-line treatment [290]. Preliminary data from the phase II INTR@PID BTC 047 study (NCT03833661) showed an ORR of 10.1% (95% CI, 5.9–15.8). This study enrolled 159 patients with advanced CCA who received bintrafusp alfa monotherapy in the second-line setting. Further clinical trials investigating the bispecific antibody bintrafusp alfa are reported in Table 28.

### 4.18. Therapy with Chimeric Antigen Receptor (CAR)-Engineered T Cells and Tumor Vaccination

Chimeric antigen receptor (CAR)-engineered T cells (CAR-T cells) have been investigated in two distinct phase I trials involving patients with advanced CCA characterized by HER2 overexpression (NCT01935843) or EGFR overexpression (NCT01869166) [291,292].

In the CART-HER-2 trial (NCT01935843), 11 patients were enrolled and received 1 to 2-cycle infusions of CART-HER2 cells (median CAR+ T cell 2.1 × 10^6^/kg). The mPFS was 4.8 months, with one patient exhibiting a partial tumor response and five patients showing stable diseases [291]. In the CART-EGFR trial (NCT01869166), 19 patients were enrolled and underwent one to three cycles of CART-EGFR cell infusion (median CART cell dose, 2.65 × 10^6^/kg; range, 0.8–4.1 × 10^6^/kg). The mPFS was 4.0 months, with one patient achieving a complete response and 10 patients demonstrating stable disease [292]. Therapy with CAR-T cells was well-tolerated in both phase I trials. However, further phase II studies are necessary to comprehensively assess the significance and efficacy of CAR-T cell therapy in CCA (see Table 29).

Data regarding tumor vaccination in patients with advanced CCA remains limited. Phase I studies focused on single vaccinations targeting MUC-1 or WT-1 in patients with advanced CCA but failed to demonstrate significant antitumor activity [293,294]. In a phase I trial utilizing a multi-target vaccine (three-peptide: cell division cycle associated 1 CDCA1, cadherin 3 CDH3, kinesin family member 20 A KIF20A), 9 enrolled patients demonstrated well-tolerated therapy and a peptide-specific T cell immune response. The trial reported an mPFS of 3.4 months and mOS of 9.7 months, with 5 out of 9 patients showing stable disease [295]. Another phase I trial employing a multi-target vaccine (four-peptide; lymphocyte antigen 6 complex locus K, TTK protein kinase, insulin-like growth factor II mRNA binding protein 3, DEP domain containing 1) enrolled 9 patients. The study reported an mPFS of 156 days and an mOS of 380 days, with a clinical response observed in 6 patients (ORR 66%) [296].

As of our knowledge, only one randomized phase II trial has compared the efficacy of chemotherapy (cyclophosphamide) alone versus chemotherapy (cyclophosphamide) combined with personalized peptide vaccination. Cyclophosphamide was administered to enhance the antigen-specific immune response. Results indicated a potential benefit for the combination therapy, with mPFS of 6.1 vs. 2.9 months and mOS of 12.1 vs. 5.9 months, favoring the combination approach [297].

Overall, the significance of tumor vaccination in CCA patients remains uncertain due to a lack of comprehensive clinical studies. Critical gaps exist, notably in understanding which patient populations could derive benefits from tumor vaccination and the efficacy of combining it with other therapies, such as chemotherapy. The current research inadequately explores these aspects. Further studies are imperative to elucidate the role and importance of tumor vaccination specifically in patients with advanced CCA. Research endeavors should aim to address these crucial gaps and provide a clearer understanding of the potential and limitations of tumor vaccination in this context.

### 4.19. Targeting Non-Coding RNA in Treatment of CCA

As reported above, non-coding RNAs have shown a potential new significance in the diagnosis as well as a predictive marker in patients with CCA. Due to their interaction with cellular processes and signaling cascades, they also hold promise as therapeutically exploitable targets. BAP1, functioning as a chromatin regulator, has been found to regulate the expression of lncRNA NEAT-1, impacting gemcitabine sensitivity in CCA cells through epigenetic mechanisms. DNA damage repair gene mutations are frequently found in CCA, and it is known that PARP inhibition results in cancer cell death via the synthetic lethality mechanism. Parasramka et al., discovered that olaparib exhibited synergistic effects with gemcitabine in CCA cells, potentially enhancing sensitivity to gemcitabine [298]. Additionally, LINC00665 was found to be significantly expressed in gemcitabine-resistant CCA cell lines and correlated with poor prognosis in CCA patients. Furthermore, the downregulation of LINC00665 reduced drug resistance in gemcitabine-resistant CCA cells, while its overexpression increased gemcitabine resistance in CCA-sensitive cells, indicating its role in enhancing chemoresistance in CCA cells [299]. Another study by Lu et al., reported that Circ-SMARCA5 expression was decreased in tumor tissues, while its elevation enhanced the sensitivity of cisplatin and gemcitabine in iCCA cells [300].

In summary, ncRNAs not only serve as biomarkers but also represent potential therapeutic targets or agents. Strategies aimed at modulating the expression or activity of disease-associated ncRNAs hold promise for the development of RNA-based therapeutics. Nonetheless, extensive clinical validation is necessary before implementing ncRNA treatment approaches in clinical practice.

## 5. Mechanisms of Therapy Resistance in CCA

Systemic treatment of CCA presents several challenges in therapeutic efficacy due to the complex interplay of mechanisms involved in chemotherapy resistance. Mechanisms of chemotherapy resistance are inherently present in healthy cholangiocytes, where they serve to defend against toxic compounds originating from the blood and bile. For this, various mechanisms are known, especially reduced drug uptake through solute carrier (e.g., via OATP1A2, OCT3, ENT1, CNT1, CTR1), increased intracellular metabolism through decreased prodrug activation or enhanced inactivation of active agents (e.g., via thymidine phosphorylase, uridine phosphorylase 1, uridine monophosphatate synthetase), or enhanced export from the cells through members of the ATB- binding cassette (ABC) superfamily (e.g., MRP1, MRP3-5, MDR1) [301]. The compensatory upregulation of molecular target structures, such as EGFR or IGF1R, for example, leads to reduced sensitivity to erlotinib (EGFR inhibitor) or substances targeting the VEGF signaling pathway (IGF1R contributes to tumor angiogenesis through upregulation of VEGF) [302,303,304]. with the introduction of targeted therapies using FGFR2 inhibitors in patients with FGFR2-fusion, secondary resistance mechanisms often emerged during treatment, including within the kinase domain of the FGFR2 receptor. with the development of newer FGFR inhibitors (including RLY-4008, tinengotinib, erdafitinib, futibatinib), which are capable of overcoming resistance mechanisms, progress has recently been made in the treatment of patients with secondary resistance mechanisms. The development of highly selective inhibitors or targeting alternative binding sites on the FGFR receptor are key structures of current research. The development of tinengotinib, a next-generation FGFR1-3 inhibitor that binds with high affinity to the active configuration of the receptor, unlike older compounds that dock at the ATP binding site, allows tinengotinib to remain effective even in the presence of resistance mutations in this domain. [124,136,137]. Various DNA repair mechanisms enable cancer cells to address different types of DNA damage caused by drugs. For instance, in 5-fluorouracil-resistant CCA cells, upregulation of uracil-DNA glycosylase activates base-excision repair [305]. Additionally, DNA excision repair protein ERCC-1 (ERCC1) participates in removing DNA adducts, and RAD51, which is elevated in most CCA, plays a role in repairing DNA double-strand breaks [306,307]. Downregulation of MutS and MutLa protein complexes involved in DNA mismatch repair leads to genetic instability, poorer prognosis, and increased chemoresistance in CCA compared to tumors without MutS and MutLa downregulation [308].

A key role in the development of therapy resistance as well as tumor progression is also played by the tumor microenvironment (TME). Macrophages within the TME, specifically tumor-associated macrophages (TAMs), display varied phenotypes and functional characteristics. CCA is distinguished by its highly desmoplastic and hypovascularized microenvironment. Previous studies have highlighted the prognostic and clinical importance of TAMs in CCA, with TAM presence correlating with unfavorable prognosis and poor survival outcomes [309]. Furthermore, studies have shown that epithelial-mesenchymal transition (EMT), characterized by the transformation of differentiated epithelial cells into a mesenchymal phenotype, is primarily linked to chemoresistance, and changes in the immune microenvironment [310].

Yang et al., reported a positive feedback loop between alternative activated macrophages (M2) and cancer cells that promotes CCA progression and chemoresistance. The authors reported that macrophages (M2) release TGFβ1, triggering the cancer cell EMT and chemoresistance via the atypical protein kinase C iota-NF-κB signaling pathway. They also demonstrated that the co-delivery of protein kinase C iota-siRNA and gemcitabine by liposomes exhibits enhanced anti-tumor effects in vitro and in vivo [309].

The precise molecular pathological mechanisms between TAMs, EMT, and tumor cells in CCA are not yet fully understood. Further research regarding these mechanisms in the future is crucial for improving CCA treatment outcomes.

## 6. Conclusions

In patients with unresectable CCA, first-line therapy with gemcitabine/cisplatin plus durvalumab or pembrolizumab is the preferred choice, non-biomarker-stratified, and irrespective of PD1/PD-L1 status.

In the case of contraindications for immunotherapy, gemcitabine/cisplatin remains the standard. The addition of durvalumab (TOPAZ-1) or pembrolizumab (KEYNOTE-966) improved mOS in patients with cholangiocarcinoma regardless of PD-L1 status. For second-line therapy, FOLFOX (ABC-06) or FOLFIRI are available for patients without molecular “druggable” targets. Druggable mutations are frequent in CCA, and early molecular testing is, thus, recommended, including NGS sequencing.

Molecular-directed therapy in cholangiocarcinoma represents a significant advancement in cancer treatment, offering more precise and personalized approaches compared to classic cytostatic chemotherapy. Molecular-directed therapies focus on inhibiting specific molecular pathways that are crucial for cancer cell survival and proliferation. In cholangiocarcinoma, pathways like FGFR2, HER2, EGFR, VEGF, and several other signaling pathways have been targeted due to their frequent dysregulation in this cancer type.

While some targeted therapies have shown promising results in clinical trials, their efficacy varies among patients. Response rates may be influenced by the presence of specific genetic mutations or alterations, necessitating comprehensive molecular profiling to identify candidates who are most likely to benefit.

Several targeted therapies have received approval for cholangiocarcinoma treatment (see Table 30), including inhibitors of FGFR (futibatinib, pemigatinib, and infigratinib), IDH1 (ivosidenib), and MSI-high/TMB-high (pembrolizumab), and NTRK (larotrectinib, entrectinib). A timeline of approved treatment options for cholangiocarcinoma is shown in Figure 5. These approvals signify a shift toward more tailored treatment options for patients. Targeted therapy is currently approved for second-line therapy in the palliative setting. Its use in first-line treatment or even in the adjuvant or neoadjuvant setting is the subject of current studies but is not currently approved. Use outside of the scope of approval should, therefore, only take place within the framework of clinical studies.

The development of new substances for the treatment of advanced CCA has progressed rapidly in the last few years. The timeline of new approvals (by FDA) for both molecular-stratified and biomarker-independent systemic therapies is demonstrated in Figure 3. Progress has also been made in the treatment of tumors with alterations in DDR genes, the VEGF signaling pathway, and the Wnt/β-catenin signaling pathway. There has also been recent progress in the field of CAR-T cell therapy. The near future will show to what extent the respective substances will reach approval and clinical routines.

Despite considerable progress in the diagnosis and treatment of CCA in recent years, there are still numerous challenges and limitations at present. The diagnosis, especially of extrahepatic CCA, continues to pose a challenge for clinicians. In attempts to histologically confirm through EUS, ERCP, or cholangioscopy, the challenge arises in obtaining adequate amounts of tumor material for diagnosis and supplementary molecular analyses. One main reason for this is the desmoplastic nature of these tumors. Given the significant molecular disparities between iCCA, eCCA, and GB, it appears justified to treat them as distinct entities, warranting separate clinical trials. Intrahepatic CCA typically exhibits longer overall survival per stage and frequently harbors identifiable driver mutations like FGFR fusions or IDH1 mutations. Conversely, extrahepatic CCA exhibits genomic resemblances to pancreatic adenocarcinoma. Another limitation is the lack of comparability of substances with similar molecular target structures. Studies in which active substances with similar molecular targets are directly compared with each other are currently lacking. Furthermore, there is currently a lack of results from studies investigating the use of molecularly targeted therapy in first-line therapy. In this regard, the results of current recruiting studies remain to be seen. Another unresolved issue is the question of the right sequential therapy, on the one hand, to prevent resistance mechanisms, and on the other hand, to respond adequately to secondarily acquired resistance mechanisms.

Another limitation of the treatment is that CCA often shows resistance to conventional chemotherapy agents, leading to poor treatment outcomes and disease progression. Furthermore, the tumor microenvironment in CCA is immunosuppressive, limiting the effectiveness of immunotherapy approaches. Future research must, therefore, also address the mechanisms of chemoresistance as well as the mechanisms of the immunosuppressive tumor microenvironment, including strategies to overcome them.

Since CCA is primarily diagnosed in advanced tumor stages, future clinical inquiries must also address the possibility of adequate downstaging procedures, aiming for secondary resectability.

Additionally, the identification of specific mutations and “biomarker”-directed targeted therapy is not effective for all patients, highlighting the importance of ongoing research to identify alternative targets and/or combination strategies. So, research into treatment options for patients without molecular alterations must also be driven forward.

The near future of molecular-directed therapy in cholangiocarcinoma will focus on the identification of additional molecular targets, refining existing therapies, and exploring combination approaches. Emerging areas of interest include targets like KRAS, DDR genes, CDK4/6, and the Wnt/β-catenin signaling pathway.

Molecular-directed therapy exemplifies the principles of personalized medicine, where treatment decisions are guided by the specific molecular characteristics of each patient’s tumor characteristics. This approach aims to improve treatment outcomes while minimizing unnecessary toxicity.

Further research is needed to address existing challenges and optimize therapeutic strategies for improved patient outcomes.

## Figures and Tables

**Figure 1 cancers-16-01690-f001:**
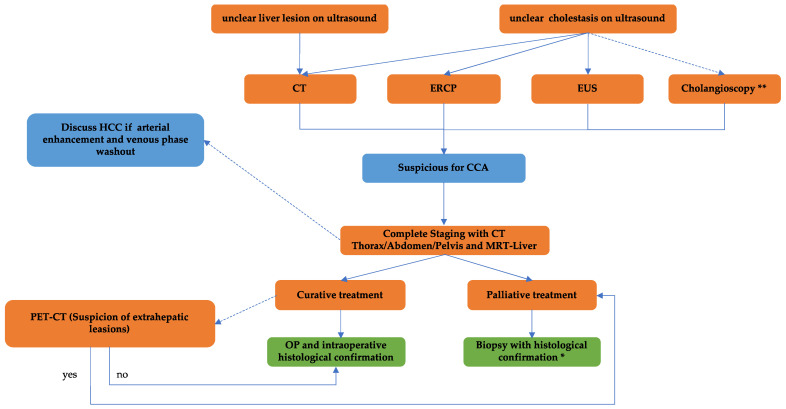
The diagnostic algorithm for cholangiocarcinoma was modified according to EASL guidelines (European Association for the Study of the Liver), NCCN guidelines (National Comprehensive Cancer Network), and S3-Leitlinie Diagnostik und Therapie des Hepatozellulären Karzinoms und biliärer Karzinome (German guidelines); * histologic confirmation via ERCP, EUS, cholangioscopy, or liver biopsy, ** if locally available.

**Figure 2 cancers-16-01690-f002:**
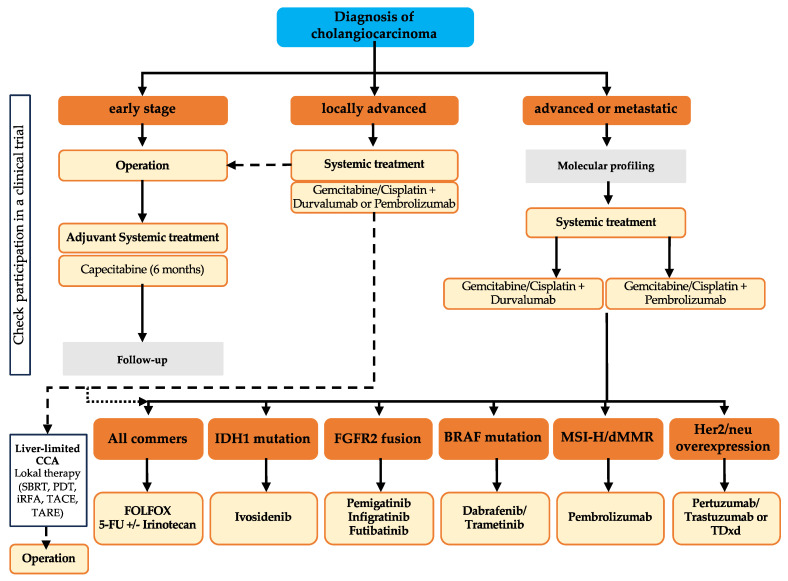
Treatment algorithm for CCA [9,72]. SBRT—stereotactic body radiotherapy; PDT—photodynamic therapy; iRFA—intraductal radiofrequency ablation; TACE—transarterial chemoembolization; TARE—transarterial radioembolization; TDxd—trastuzumab deruxtecan. The dashed arrows denote individual decision-making in the treatment of locally advanced CCA as a neoadjuvant treatment or conversion therapy strategy. Pointed arrows define the pathway if neoadjuvant treatment fails.

**Figure 3 cancers-16-01690-f003:**
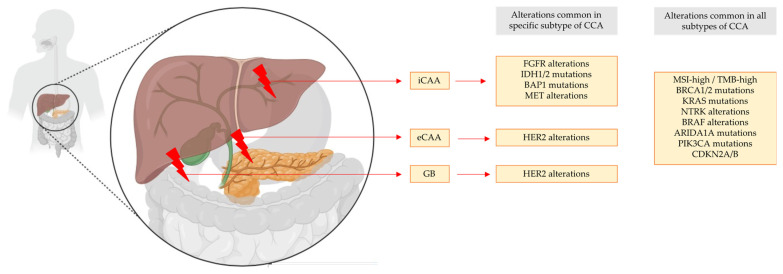
Most frequent and currently druggable molecular alterations by CCA subtype (notably, for the approval status of the potential agents, see the section below). Created with Biorender.

**Figure 4 cancers-16-01690-f004:**
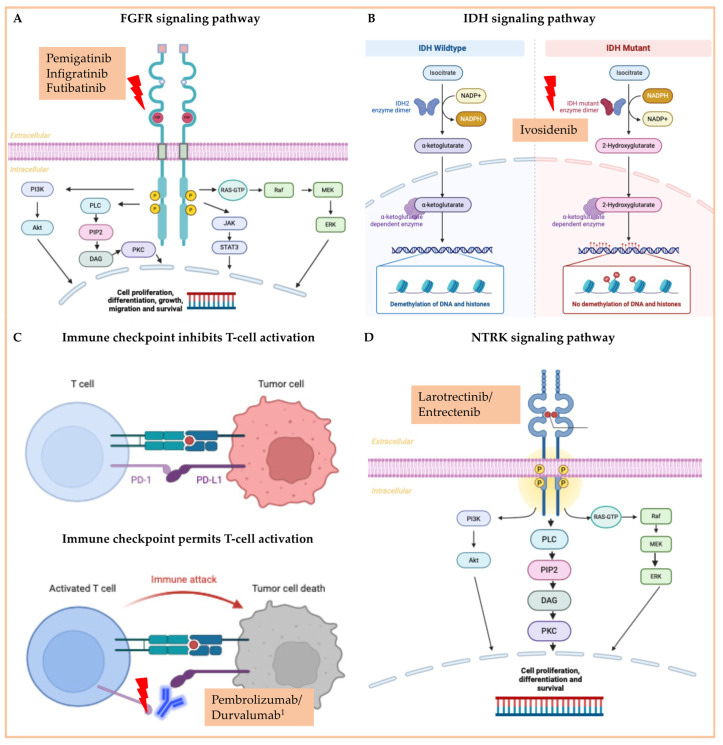
Key signaling pathways for FDA-approved molecular-directed drugs for the treatment of advanced CCA in the palliative setting. (**A**) FGFR signaling pathway. (**B**) IDH signaling pathway. (**C**) Mechanism of immune checkpoint inhibition. (**D**) NTRK signaling pathway. ^1^ Durvalumab/pembrolizumab in combination with gemcitabine/cisplatin for all-comers, pembrolizumab monotherapy for MSI-high/TMB-high. Created and modified with Biorender.

**Figure 5 cancers-16-01690-f005:**
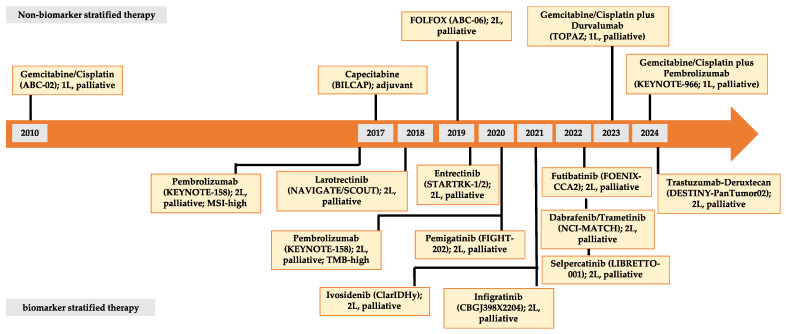
Timeline: targeted therapy with FDA approval for treatment of advanced cholangiocarcinoma. 1L: first line; 2L: second line.

**Table 1 cancers-16-01690-t001:** Risk factors for iCCA and eCCA [10].

Risk Factor	Study Design	Risk Estimate (95% CI)	References
Alcohol abuse	Case–control	iCCA: 5.9 (2.1–17.4); 7.4 (4.3–12.8); 3.1 (1.3–7.5) eCCA: 3.6 (1.5–9.4); 4.5 (2.2–9.1)	[7,11,12,13,14]
	Case–control	iCCA: 3.72 (3.17–4.35) eCCA: 2.60 (2.23–3.04)	
	Meta-analysis	2.81 (1.52–5.21)	
	Pooled analysis	iCCA: 2.35 (1.46–3.78) eCCA: 1.82 (0.98–3.39)	
Asbestos	Cohort	1.50 (1.16–1.94)	[15]
Bile duct cysts	Case–control	iCCA: 15.66 (11.58–21.18); 36.9 (22.7–59.7) eCCA: 27.12 (22.06–33.34); 47.1 (30.4–73.2)	[14,16,17,18]
Caroli’s disease	Case-control	iCCA: 38.13 (14.20–102.38) eCCA: 96.81 (51.02–183.68)	[14,19]
Cholangitis	Case–control	iCCA: 21.52 (17.21–26.90); 6.32 (2.3–17.5); 8.8 (4.9–16.0); 64.2 (47.7–86.5) eCCA: 40.80 (34.96–47.60); 45.7 (32.9–63.3)	[14,18,20,21]
Cholelithiasis	Case–control	iCCA: 3.93 (3.49–4.43) eCCA: 5.29 (4.83–5.80)	[7,13,14]
Choledocholithiasis	Case–control	iCCA: 6.94 (5.64–8.54); 23.97 (2.9–198.9); 4.0 (1.9–8.5); 22.5 (16.9–30.0) eCCA: 14.22 (12.48–16.20); 34.0 (26.6–43.6)	[13,14,18,20,21]
Chronic inflammatory bowel disease	Case–control	iCCA: 4.67 (1.6–13.9)	[18,22]
Chronic hepatitis B viral infection	Case–control	iCCA: 2.97 (1.97–4.46); 28.6 (3.9–1268.1); 0.8 (0.1–5.9); 2.7 (0.4–18.5); 2.3 (1.6–3.3); 8.9 (5.97–13.2) eCCA: 2.38 (1.65–3.44); 3.2 (0.6–382)	[7,11,14,20,23,24,25,26]
Chronic hepatitis C viral infection	Case–control	iCCA: 4.67 (3.57–6.11); 6.02 (1.5–24.1); 7.9 (1.3–84.5); 5.2 (2.1–12.8); 2.2 (1.4–14.0); 9.7 (1.6–58.9); 0.93 (0.3–3.1) eCCA: 3.18 (2.43–4.16); 2.8 (0.3–35.1); 1–5 (0.2–11.0)	[7,11,14,18,20,23,24,26,27,28]
	Cohort	iCCA: 2.55 (1.3–4.9) eCCA: 1.05 (0.6–1.9)	
Chronic pancreatitis	Case–control	iCCA: 2.66 (1.72–4.10) eCCA: 6.61 (5.21–8.40)	[14,29]
Cirrhosis of the liver	Case–control	iCCA: 5.03 (0.045–56.82); 27.2 (19.9–37.1); 10.0 (6.1–16.4); 13.6 (6.5–28.5) eCCA: 5.4 (2.9–10.2)	[7,11,18,20,25,28,30]
Crohn’s disease	Case–control	iCCA: 1.77 (1.13–2.75); 2.0 (0.6–6.3); 2.4 (1.0–5.9) eCCA: 1.71 (1.17–2.51); 2.8 (1.3–6.4	[14,18,20,31]
	Cohort	iCCA/eCCA: 3.0 (0.9–8.6)	
Diabetes mellitus type 1	Case–control	iCCA: 1.43 (1.25–1.63) eCCA: 1.30 (1.16–1.46)	[7,11,14,32,33]
Diabetes mellitus type 2	Case–control	iCCA: 1.54 (1.41–1.68) eCCA: 1.45 (1.34–1.56)	[7,11,14,32,33]
Duodenal/gastric ulcer	Case–control	iCCA: 1.14 (1.21–1.66) eCCA: 1.46 (1.29–1.66)	[14]
Hemochromatosis	Case–control	iCCA: 2.07 (1.33–3.22)	[14,34,35,36]
liver flukes	N/A	N/A	[37,38,39]
Non-alcoholic fatty liver disease	Case–control	iCCA: 3.52 (2.87–4.32) eCCA: 2.93 (2.42–3.55)	[7,14,40]
Obesity	Case–control	iCCA: 1.42 (1.21–1.66); 1.7 (1.1–2.6); 2.05 (0.7–5.6) eCCA: 1.17 (1.01–1.35); 1.1 (0.7–1.8)	[11,14,18,21,32]
Primary biliary cholangitis	Case–control	iCCA: 9.84 (6.24–15.52) eCCA: 8.34 (5.44–12.78)	[14]
Primary sclerosing cholangitis	Case–control	iCCA 93.4 (27.1–322) distal eCCA 34.0 (3.6–323) perihilar eCCA 453 (104–999)	[7,41,42,43,44,45]
Smoking	Case–control	iCCA: 1.46 (1.28–1.66); 1.8 (1.0–3.2); 1.8 (1.2–2.7) eCCA: 1.77 (1.59–1.96); 1.7 (1.0–3.0)	[7,12,14,18,23,46]
Thorotrast	N/A	N/A	[47]
Ulcerative colitis	Case–control	iCCA: 2.18 (1.61–2.95); 2.2 (1.2–3.9); 4.5 (2.6–7.9) eCCA: 1.75 (1.32–2.33); 1.7 (0.7–4.0)	[14,18,20,31]
	Cohort	iCCA/eCCA: 4.1 (2.4–6.8)	

eCCA: extrahepatic cholangiocarcinoma; iCCA: intrahepatic cholangiocarcinoma. N/A = not available.

**Table 2 cancers-16-01690-t002:** Risk factors for GB [10].

Risk Factor	Study Design	Risk Estimate (95% CI)	References
Aflatoxin	Case–control	2.0 (1.0–3.9)	[48,49]
Age	N/A	N/A	[48]
Anatomical anomalies of the intra- and extrahepatic bile ducts	N/A	N/A	[48]
Arsen	Retrospective analysis	1.72 (1.54–1.91); 1.45 (1.30–1.62)	[48,50]
Cholecystolithiasis	Case–control	5.3 (1.5–18.9)	[48,51,52]
Crohn’s disease	Case–control	1.83 (1.23–2.71)	[48]
Diabetes mellitus	Case–control	2.7 (1.2–6.4)	[48,52]
Female sex	Case–control	2.4 (1.3–4.3)	[48,52]
Gallbladder polyps	Retrospective analysis	8.147 (2.56–23.40	[48,51,53]
Liver flukes/infections	N/A	Helicobacter pilis OR of 6.5 in Japanese patients and 5.86 in Thai patients; S. typhi carriers have an 8 to 12-fold increased risk of developing GB	[48,54]
Obesity	Meta-analysis	1.10 (1.02–1.1); 1.69 (1.54–1.86)	[48,51]
Porcelain gallbladder	Retrospective analysis	8.0 (1.0–63.0); if symptoms 83.6 (2.3–2979.1); if gallbladder mass 3226.6 (17.2–603884.8)	[48,55,56]
Primary sclerosing cholangitis	Case–control	2.06 (1.27–3.33)	[48,51]
Tobacco	Case–control	3.8 (1.7–8.1)	[52]

**Table 3 cancers-16-01690-t003:** Prevalence of druggable targets in CCA [103,104,105].

Target	iCCA	eCCA	GB
FGFR2 alterration	9–13%	0%	2–7%
IDH1/2 mutation	10–29%	3–5%	0–2%
BRAF mutation	5%	2–3%	0–1%
HER2/ERBB2 overexpression	3–8%	1.3–11%	6–15%
KRAS mutation	15–22%	38–57%	7–10%
ARID1A mutation	18–23%	12–20%	12–17%
PIK3CA mutation	3–7%	5–7%	9–10%
CDKN2A/B mutation	9–27%	9–28%	12–25%
MET overexpression	2–4%	0%	1–2%
BAP1 mutation	15–19%	0%	3–13%
RET fusion	0–5%	0–5%	0–5%
NRTK fusion	<1–2%	<1–2%	<1–2%
MSI-high	Up to 5%	Up to 5%	Up to 5%
BRCA1/2 mutation	Up to 3–5%	Up to 3–5%	Up to 3–5%
EGFR overexpression	11–27%	5–19%	N/A

**Table 4 cancers-16-01690-t004:** Ongoing studies for pemigatinib in CCA.

Study	Phase	Setting	Treatment Arm A	Treatment Arm B	Primary Endpoint	NCT
FIGHT-302	III	1L Palliative	Pemigatinib	Gemcitabine/cisplatin	PFS	NCT03656536
PEARLDIFER	II	Adjuvant	Pemigatinib	-	ORR	NCT05565794
-	II	1L Palliative	Pemigatinib plus sintilimab	-	ORR	NCT05913661
-	I	Palliative	Gemcitabine/cisplatin plus pemigatinib (FGFR2 alteration)	Gemcitabine/cisplatin plus Ivosidenib (IDH1 mutation)	safety	NCT04088188

1L: first-line; ORR: overall response rate; PFS: progression-free survival.

**Table 5 cancers-16-01690-t005:** Ongoing studies for futibatinib in CCA.

Study	Phase	Setting	Treatment Arm A	Treatment Arm B	Primary Endpoint	NCT
FOENIX-CCA3	III	1L Palliative	Futibatinib	Gemcitabine/cisplatin	PFS	NCT04093362
FOENIX-CCA4	II	Palliative	Futibatinib (20 mg)	Futibatinib (16 mg)	ORR	NCT05727176

1L: first-line; ORR: overall response rate; PFS: progression-free survival.

**Table 6 cancers-16-01690-t006:** Ongoing studies for infigratinib in CCA.

Study	Phase	Setting	Treatment Arm A	Treatment Arm B	Primary Endpoint	NCT
OPTIC	II	Neoadjuvant	Gemcitabine/cisplatin/nab-Paclitaxel plus infigratinib	Gemcitabine/cisplatin/nab-paclitaxel	Safety	NCT05514912
-	III	1L, Palliative	Infigratinib	Gemcitabine/cisplatin	PFS	NCT03773302

1L: first-line; PFS: progression-free survival.

**Table 7 cancers-16-01690-t007:** Ongoing studies for derazantinib in CCA.

Study	Phase	Setting	Treatment Arm A	Treatment Arm B	Primary Endpoint	NCT
ADVANCE_2020	II	Palliative	Atezolizumab plus derazantinib	-	Safety	NCT05174650

**Table 8 cancers-16-01690-t008:** Ongoing studies for novel FGFR2 inhibitors in CCA.

Study	Phase	Setting	Treatment Arm A	Treatment Arm B	Primary Endpoint	NCT
ADVANCE_2020	I	Palliative	Bemarituzumab (dose-finding)	-	Safety	NCT02318329
REFOCUS	I/II	Palliative	RLY-4008	-	Safety	NCT04526106
-	II	Palliative	Tinengotinib	-	ORR	NCT06057571
FIRST-308	III	Palliative	Tinengotinib	Physician’s choice	Safety, PFS	NCT05948475

ORR: overall response rate; PFS: progression-free survival.

**Table 9 cancers-16-01690-t009:** Ongoing studies for ivosidenib in CCA.

Study	Phase	Setting	Treatment Arm A	Treatment Arm B	Primary Endpoint	NCT
ProvIDHe	III	Palliative	Ivosidenib	-	Safety	NCT05876754
-	II	Palliative	Ivosidenib plus nivolumab	-	OR	NCT04056910
-	I/II	Palliative	Ivosidenib plus nivolumab plus ipilimumab	-	Safety, OR	NCT05921760
-	I	Palliative	Gemcitabine/cisplatin plus pemigatinib (FGFR2 alteration)	Gemcitabine/cisplatin plus ivosidenib (IDH1 mutation)	Safety	NCT04088188

ORR: overall response rate.

**Table 10 cancers-16-01690-t010:** Ongoing studies targeting IDH 1/IDH 2 in CCA.

Study	Phase	Setting	Treatment Arm A	Treatment Arm B	Primary Endpoint	NCT
-	II	Palliative	Dasatinib	-	ORR	NCT02428855
-	I/II	Palliative	Olutasidenib	Olutasidenib plus azacitidine or nivolumab or gemcitabine/cisplatin	Safety, ORR	NCT03684811
-	I/II	Palliative	Enasidenib	-	Safety	NCT02273739
-	II	Palliative	BAY-1436032	-	Safety	NCT02746081
-	I	Palliative	LY3410738	LY3410738 plus gemcitabine/cisplatin plus durvalumab	Safety	NCT04521686
-	I	Palliative	IDH305	-	Safety	NCT02381886
-	I	Palliative	Vorasidenib	-	Safety	NCT02481154

ORR: overall response rate.

**Table 11 cancers-16-01690-t011:** Ongoing studies for molecular targeting HER2 in CCA.

Study	Phase	Setting	Treatment Arm A	Treatment Arm B	Primary Endpoint	NCT
-	I/II	Palliative	Zanidatamab plus evorpacept	-	Safety, ORR	NCT05027139
-	II	Palliative	Zanidatamab plus cisplatin/5-FU or mFOLFOX6 or XELOX or mFOLFOX6 plus bevacizumab or Gemcitabine/cisplatin	-	Safety, ORR	NCT03929666
-	I	Palliative	Tipifarnib plus trastuzumab	-	Safety	NCT00005842
DPT02	II	Palliative	TDxd	-	ORR	NCT04482309
-	I	Palliative	ZN-A-1041 (dose-finding)	ZN-A-1041 plus TDxd or TD-M1 or pertuzumab/trastuzumab	Safety	NCT05593094
-	II	Palliative	TD-M1	-	ORR	NCT02675829
TAPISTRY	II	Palliative	TD-M1	-	ORR	NCT04589845

ORR: overall response rate, TD-M1: trastuzumab emtansine, TDxd: trastuzumab deruxtecan.

**Table 12 cancers-16-01690-t012:** Ongoing studies for molecular targeting NTRK in CCA.

Study	Phase	Setting	Treatment Arm A	Treatment Arm B	Primary Endpoint	NCT
-	II	Palliative	Larotrectinib	-	OR	NCT04879121
TAPISTRY	II	Palliative	Entrectinib	-	Safety	NCT04589845

OR: overall response.

**Table 13 cancers-16-01690-t013:** Ongoing studies for molecular targeting BRAF in CCA.

Study	Phase	Setting	Treatment Arm A	Treatment Arm B	Primary Endpoint	NCT
-	II	Palliative	Selumetinib plus gemcitabine/cisplatin	Gemcitabine/cisplatin	Response	NCT02151084
BEAVER	II	Palliative	Binimetinib plus encorafenib	-	ORR	NCT03839342
-	I	Palliative	ABM-1310	ABM-1310 plus cobimetinib	safety	NCT04190628
TAPISTRY	II	Palliative	Belvarafenib	-	ORR	NCT04589845
-	-	Palliative	Ulixertinib	-	-	NCT04566393
-	II	Palliative	LY3214996 plus abemaciclib	-	ORR	NCT04534283

ORR: overall response rate.

**Table 14 cancers-16-01690-t014:** Ongoing studies for molecular targeting DDR genes in CCA.

Study	Phase	Setting	Treatment Arm A	Treatment Arm B	Primary Endpoint	NCT
-	I	Palliative	Niraparib plus anlotinib	-	Safety	NCT02151084
UF-STO-ETI-001	II	Palliative	Niraparib	-	ORR	NCT03207347
-	II	Palliative	Olaparib	-	PFS	NCT04042831
-	II	Palliative	Pembrolizumab plus olaparib	-	ORR	NCT04306367
-	II	Palliative	Nivolumab plus olaparib	-	Response	NCT03639935
-	I/II	Palliative	Liposomal irinotecan plus 5-FU plus rucaparib	-	Response, safety	NCT03337087
-	I	Palliative	Veliparib plus gemcitabine/cisplatin	-	Safety	NCT01282333
DDR-Umbrella	II	Palliative	Ceralasertib plus durvalumab	Ceralasertib plus Durvalumab	DCR	NCT04298021
-	II	Palliative	Rucaparib plus nivolumab	-	Response	NCT03639935
-	II	Palliative	Adavosertib	-	OR	NCT02465060
-	II	Palliative	Olaparib	-	ORR	NCT03212274
-	II	Palliative	AZD6738 plus durvalumab	-	DCR	NCT04298008
SOLID	II	Palliative	Olaparib plus durvalumab	-	ORR, DCR	NCT03991832
-	II	Palliative	Toripalimab	-	ORR	NCT03810339
-	II	Palliative	Pembrolizumab	-	Response	NCT03428802
OPTIMUM	II	Palliative	Olaparib plus durvalumab	Olaparib	PFS	NCT05222971
-	I	palliative	Copanlisib plus olaparib plus durvalumab	-	Safety	NCT03842228
TAPISTRY	II	palliative	Camonsertib	-	ORR	NCT04589845

DCR: disease control rate; OR: objective response; ORR: overall response rate; PFS: progression-free survival.

**Table 15 cancers-16-01690-t015:** Ongoing studies for molecular targeting ROS1/ALK/MET in CCA.

Study	Phase	Setting	Treatment Arm A	Treatment Arm B	Primary endpoint	NCT
TalaCom	I	Palliative	Talazoparib plus crizotinib	-	Safety	NCT04693468
STARTRK-2	II	Palliative	Entrectinib	-	ORR	NCT02568267
TAPISTRY	II	Palliative	Alectinib	-	ORR	NCT04589845
TAPISTRY	II	Palliative	Entrectinib	-	ORR	NCT04589845

ORR: overall response rate.

**Table 16 cancers-16-01690-t016:** Ongoing studies for molecular targeting Receptor Tyrosine Kinase RET.

Study	Phase	Setting	Treatment Arm A	Treatment Arm B	Primary Endpoint	NCT
TAPISTRY	II	Palliative	Pralsetinib	-	ORR	NCT04589845

ORR: overall response rate.

**Table 17 cancers-16-01690-t017:** Ongoing studies for molecular targeting PI3K/AKT/mTOR in CCA.

Study	Phase	Setting	Treatment Arm A	Treatment Arm B	Primary Endpoint	NCT
TAPISTRY	II	Palliative	Inavolisib	-	ORR	NCT04589845
TAPISTRY	II	Palliative	Ipatasertib	-	ORR	NCT04589845
-	I	Palliative	Copanlisib plus olaparib plus durvalumab	-	Safety	NCT03842228

ORR: overall response rate.

**Table 18 cancers-16-01690-t018:** Ongoing studies for molecular targeting WNT/β-catenin signaling pathway in CCA.

Study	Phase	Setting	Treatment Arm A	Treatment Arm B	Primary Endpoint	NCT
KEYNOTE 596	I	Palliative	CGX1321	-	Safety	NCT02675946
-	I	Palliative	DKN-01 plus gemcitabine/cisplatin	-	Safety	NCT02375880
-	II	Palliative	DKN-01 plus nivolumab	-	ORR	NCT04057365

ORR: overall response rate.

**Table 19 cancers-16-01690-t019:** Ongoing studies for molecular targeting CDK in CCA.

Study	Phase	Setting	Treatment Arm A	Treatment Arm B	Primary Endpoint	NCT
-	I/II	Palliative	Abemaciclib plus paclitaxel	-	Safety, ORR	NCT04594005
-	II	Palliative	Abemaciclib	-	PFR	NCT03310879

ORR: overall response rate, PFR: progression-free rate.

**Table 20 cancers-16-01690-t020:** Ongoing studies for molecular targeting RAS/RAF/MEK/ERK in CCA.

Study	Phase	Setting	Treatment Arm A	Treatment Arm B	Primary Endpoint	NCT
-	II	Palliative	LY3214996 plus abemaciclib	-	ORR	NCT04534283
-	-	Palliative	Ulixertinib	-	-	NCT04566393
TAPISTRY	II	Palliative	GDC-6036	-	ORR	NCT04589845
-	I	Palliative	LY3537982 plus abemaciclib	-	Safety	NCT04956640
-	I/II	Palliative	GFH925	-	Safety, ORR	NCT05005234
KRYSTAL-16	I	Palliative	Adagrasib plus palbociclib	-	Safety	NCT05178888
-	II	Palliative	Sotorasib plus panitumumab	-	ORR	NCT05993455
CodeBreak 101	II	Palliative	Sotorasib	-	Safety, ORR	NCT04185883
CodeBreak 101	I/II	Palliative	Sotorasib	-	Safety, ORR, DOR, TTR	NCT03600883
KRYSTAL-19	I/II	Palliative	Adagrasib plus nab-sirolimus	-	Safety, ORR	NCT05840510
-	I/II	Palliative	JAB-21822	-	Safety, ORR, DOR	NCT05002270
-	I/II	Palliative	JAB-21822 plus cetuximab	-	Safety, ORR	NCT05194995
-	I/II	Palliative	YL-15293	-	ORR	NCT05119933
-	I	Palliative	HBI-2438	-	Safety	NCT05485974
-	I/II	Palliative	HS-10370	-	Safety	NCT05367778
-	I	Palliative	MRTX849	-	Safety	NCT05263986
-	I/II	Palliative	BMS-986466 plus adagrasib	BMS-986466 plus Adagrasib plus Cetuximab	Safety, ORR	NCT06024174
-	I/II	Palliative	MRTX0902	MRTX0902 plus Adagrasib	ORR, DOR, OS, PFS	NCT05578092
-	I	Palliative	MK-1084	-	Safety	NCT05067283
KontRASt-01	I/II	Palliative	JDQ443	JDQ443 plus TNO155 or tislelizumab or both	Safety, ORR	NCT04699188
-	I	Palliative	RMC-6291	-	Safety	NCT05462717
-	I	Palliative	SY-5933	-	Safety	NCT06006793
-	I	Palliative	D3S-001	-	Safety	NCT05410145
-	I	Palliative	GEC255	-	Safety	NCT05768321
AMPLIFY-7P	I/II	Palliative	ELI-002 7P	-	Safety, DFS	NCT05726864

DFS: disease-free survival; DOR: duration of response; ORR: overall response rate; OS: overall survival; PFS: progression-free survival; TTR: time to response.

**Table 21 cancers-16-01690-t021:** Ongoing studies for bevacizumab in CCA.

Study	Phase	Setting	Treatment Arm A	Treatment Arm B	Primary Endpoint	NCT
-	II	Palliative	Sintilimab plus IBI305 plus GEMOX	Sintilimab plus GEMOX (Arm C: GEMOX)	ORR	NCT05251662
-	II	Palliative	Atezolizumab plus bevacizumab plus Gemcitabine/cisplatin	Atezolizumab plus gemcitabine/cisplatin	PFS	NCT05211323

ORR: overall response rate, PFS: progression-free survival.

**Table 22 cancers-16-01690-t022:** Ongoing studies for lenvatinib in CCA.

Study	Phase	Setting	Treatment Arm A	Treatment Arm B	Primary Endpoint	NCT
-	II	Palliative	Lenvatinib plus gemcitabine/cisplatin	-	ORR	NCT04527679
-	II	Palliative	Sintilimab plus lenvatinib	-	ORR	NCT05010681
-	-	-	GEMOX plus lenvatinib plus toripalimab	GEMOX plus toripalimab (Arm C: lenvatinib plus toripalimab)	ORR, safety	NCT05215665
-	III	Palliative	GEMOX or gemcitabine/cisplatin plus lenvatinib plus toripalimab	GEMOX or gemcitabine/cisplatin plus toripalimab (Arm C: GEMOX or gemcitabine/cisplatin)	OS	NCT05342194
-	II	Palliative	Lenvatinib plus tislelizumab plus gemcitabine/cisplatin	Gemcitabine/cisplatin	ORR	NCT05532059
-	II	Palliative	Lenvatinib plus tislelizumab plus gemcitabine/oxaliplatin	Tislelizumab plus gemcitabine/oxaliplatin	OR	NCT05620498
-	II	Palliative	Toripalimab plus lenvatinib	-	ORR, safety	NCT04211168
-	II/III	Neoadjuvant	Toripalimab plus lenvatinib plus GEMOX	-	Event-free survival	NCT04669496

ORR: overall response rate, OS: overall survival.

**Table 23 cancers-16-01690-t023:** Ongoing studies for regorafenib in CCA.

Study	Phase	Setting	Treatment Arm A	Treatment Arm B	Primary Endpoint	NCT
BREGO	II	Palliative	Regorafenib plus GEMOX	GEMOX	Safety	NCT02386397

**Table 24 cancers-16-01690-t024:** Ongoing studies for anlotinib in CCA.

Study	Phase	Setting	Treatment Arm A	Treatment Arm B	Primary Endpoint	NCT
-	II	Palliative	Anlotinib plus sintilimab plus GEMOX	-	ORR	NCT06033118

ORR: overall response rate.

**Table 25 cancers-16-01690-t025:** Ongoing studies for CPI monotherapy in CCA.

Study	Phase	Setting	Treatment Arm A	Treatment Arm B	Primary Endpoint	NCT
KEYNOTE-158	II	Palliative	Pembrolizumab	-	ORR	NCT02628067

ORR: overall response rate.

**Table 26 cancers-16-01690-t026:** Ongoing studies for CPI combination in CCA.

Study	Phase	Setting	Treatment Arm A	Treatment Arm B	Primary Endpoint	NCT
DART	II	Palliative	Nivolumab plus ipilimumab	Nivolumab	ORR	NCT02834013
-	II	Palliative	Nivolumab plus ipilimumab	-	Efficacy	NCT02923934
IMMUNO-BIL	II	Palliative	Durvalumab plus tremelimumab	-	OS	NCT03704480

ORR: overall response rate.

**Table 27 cancers-16-01690-t027:** Ongoing studies for CPI in combination with cytostatic chemotherapy or other agents in CCA.

Study	Phase	Setting	Treatment Arm A	Treatment Arm B	Primary Endpoint	NCT
-	II	Palliative	Pembrolizumab plus GC	-	PFS	NCT03260712
-	II	Perioperative	Pembrolizumab plus GC	-	Safety	NCT05967182
-	II	Palliative	Toripalimab plus GC	-	PFS, OS	NCT03796429
-	II	Palliative	Toripalimab plus gemcitabine plus 5-FU	-	Safety, PFS	NCT03982680
-	II	Palliative	Toripalimab plus S1 plus nab-paclitaxel	-	ORR	NCT04027764
-	II	Palliative	Lenvatinib plus toripalimab	Lenvatinib plus toripalimab plus GEMOX (Arm C: GEMOX plus toripalimab)	ORR, safety	NCT05215665
-	III	Palliative	Lenvatinib plus toripalimab plus GC or GEMOX	Toripalimab plus GC or GEMOX	OS	NCT05342194
-	II	Palliative	Lenvatinib plus toripalimab	-	ORR, safety	NCT04211168
-	II/III	Neoadjuvant	Lenvatinib plus toripalimab plus GEMOX	-	Event free survival	NCT04669496
-	II	Palliative	Axitinib plus toripalimab	-	ORR, PFS	NCT04010071
-	III	Palliative	Envafolimab plus GEMOX	-	OS	NCT03478488
DEBATE	II	Neoadjuvant	Durvalumab plus GC	GC	R0 resection	NCT04308174
-	II	Neoadjuvant	Durvalumab plus GC	-	Relapse-free survival rate	NCT05672537
-	II	Neoadjuvant	Durvalumab plus GC	-	Complete treatment	NCT06050252
-	II	Neoadjuvant	Durvalumab plus tremelimumab plus GC	-	Conversion from unresectable to resectable	NCT06017297
-	I/II	Neoadjuvant	Durvalumab plus tremelimumab plus GC	-	ORR	NCT04989218
ADJUBIL	II	Adjuvant	Durvalumab plus tremelimumab plus capecitabine	Durvalumab plus tremelimumab	RFS	NCT05239169
NeoTreme	II	Neoadjuvant	Durvalumab plus tremelimumab plus GC	-	-	-
-	I	Palliative	Durvalumab plus guadecitabine	-	Safety, response	NCT03257761
BATTALION	II	Palliative	Botensilimab plus balstilimab plus GC	-	ORR	-
-	II	Palliative	Atezolizumab	Atezolizumab plus cobimetinib	PFS	NCT03201458
-	II	2L, Palliative	SHR1316 plus IBI310	-	ORR	NCT04634058

2L: second-line; GC: Gemcitabine/Cisplatin; GEMOX: Gemcitabine/Oxaliplatin; ORR: overall response rate, OS: overall survival; PFS: progression-free survival; RFS: recurrence free survival.

**Table 28 cancers-16-01690-t028:** Ongoing studies for bintrafusp alfa in CCA.

Study	Phase	Setting	Treatment Arm A	Treatment Arm B	Primary Endpoint	NCT
-	II/III	Palliative	Bintrafusp alfa plus gemcitabine/cisplatin	Gemcitabine/cisplatin	Safety, OS	NCT04066491
-	II	Palliative	Bintrafusp alfa	-	Response	NCT03833661

OS: overall survival.

**Table 29 cancers-16-01690-t029:** Ongoing studies for CART-cells in CCA.

Study	Phase	Setting	Treatment Arm A	Treatment Arm B	Primary Endpoint	NCT
-	I/II	Palliative	MUC-1 CART	-	DCR	NCT03633773

DCR: disease control rate.

**Table 30 cancers-16-01690-t030:** Approved molecular-directed therapies in CCA.

Study	Phase	Setting	Drug	Target	FDA	EMA	NCT
FIGHT-202	III	2L, Palliative	Pemigatinib	FGFR	x	x	NCT02924376
FOENIX-CCA2	II	2L, Palliative	Futibatinib	FGFR	x	x	NCT02052778
CBGJ398X2204	II	2L, Palliative	Infigratinib	FGFR	x	-	NCT02150967
Keynote-158	II	2L, Palliative	Pembrolizumab	MSI-high	x	x	NCT02628067
Keynote-158	II	2L, Palliative	Pembrolizumab	TMB-high	x	-	NCT02628067
ClarIDHy	III	2L, Palliative	Ivosidenib	IDH1	x	x	NCT02989857
STARTRK-1/2	I/II	2L, Palliative	Entrectinib	NTRK	x	x	NCT02097810 NCT02568267
NAVIGATE SCOUT	I/II	2L, Palliative	Larotrectinib	NTRK	x	x	NCT02576431 NCT02637687
NCI-MATCH	II	2L, Palliative	Dabrafenib and Trametinib	BRAFV600E	x	-	NCT03155620
LIBRETTO-001	I/II	2L, Palliative	Selpercatinib	RET	x	-	NCT03157128
DESTINY-PanTumor02	II	2L, Palliative	Trastuzumab–deruxtecan	HER2^+^	x	-	NCT04482309

1L: first-line; 2L: second-line; ^+^ only IHC 3+.

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
