# Peer review of "Current and Future Therapeutic Targets for Directed Molecular Therapies in Cholangiocarcinoma"

_cancers, 2024, doi:10.3390/cancers16091690_

Round 1

Reviewer 1 Report

Comments and Suggestions for Authors

This is an outstanding review on the ongoing efforts to bring basic science evidence to the bedside of patients with biliary tract malignancies. I have only minor observations:

1) Table 1 and 2 benefit from more information. Please add the basic study design and the main outcome of such study (OR, RR, HR), that way the reader can have a better idea of how important the risk factor is.

2) The manuscript is necessarily long. Having said that, the "Diagnostic work-up" section offers little contribution to the study and I believe it could be eliminated along with Figure 1 without any detriment to the manuscript.

3) Please a Figure illustrating the location of the proteins mutated in the cell of each type of CCA (extrahepatic, perihilar and intrahepatic). Several mutations have been described in all CCA but some are type-specific. This way the reader can understand visually the process that the targeted drug is blocking in the cell.

4) Please add in the Conclusion section what is already mentioned in Table 30. At the present time, these targeted therapies have been approved as second-line or palliative therapies. Given the current excitement about them, some clinicians may misplace them in our current practice, something that should be reserved for clinical trials only.

Congratulations to the authors on this great effort.  

Author Response

Dear Reviewer,
Thank you very much for your valuable and comprehensive suggestions for corrections. Please find attached the revised manuscript and the point-by-point response. Please see the attachment.
Yours sincerely
Dr. Philipp Heumann and Prof. Dr. Arne Kandulski.

Reviewer 2 Report

Comments and Suggestions for Authors

The article you provided discusses various aspects of cholangiocarcinoma (CCA), including imaging approaches, diagnostic methods, staging, treatment options, and ongoing studies on molecular targeting in CCA.

Stage-dependent therapeutic regimes

Discuss the role of systemic therapy, such as gemcitabine/cisplatin and immunotherapy, in the palliative setting.

Author Response

(The authors gave the same response as above.)

Reviewer 3 Report

Comments and Suggestions for Authors

The present review highlights the clinical trials and palliative treatment regimes used as treatment options for cholangiocarcinoma. besides, they provide details about the trials ongoing and recruited for targeting FGFR-2, BRAF, EGFR etc. However, many critical points need to be addressed.

Major points:

Ø  The review lacks a discussion about the crosstalk of molecular targets among the HCC, iCCA and GB.

Ø  It also lacks a description of the negative feedback loop disrupting the therapeutic efficacy in CCA and is responsible for resistance.

Ø  The review lacks a discussion about the noncoding RNA that serves as molecular biomarkers and therapeutic targets for CCA.

Ø  Review the lack of in-depth discussion about the current therapeutic challenges/ limitations.

Minor point:

Ø  A unified table along with all the clinical trials addressing different molecular targets would be more appropriate.

Ø  A schematic figure representing molecular and therapeutic targets along with the available treatment will help to understand the current therapeutic scenario of cholangiocarcinoma.

Ø  A figure describing the timeline advancements in therapeutic will be helpful.

Author Response

(The authors gave the same response as above.)
